# Automated Formalization via Conceptual Retrieval-Augmented LLMs

**Wangyue Lu**[1][*]   **Lun Du**[2][*]   **Sirui Li**[1]   **Ke Weng**[1]   **Haozhe Sun**[1]   **Hengyu Liu**[3][†]

**Minghe Yu**[4]   **Tiancheng Zhang**[1][†]   **Ge Yu**[1]

[1]School of Computer Science and Engineering, Northeastern University, Shenyang 110819, China
[2]Ant Research, Ant Group, Beijing, China
[3]Department of Computer Science, Aalborg University, Denmark
[4]Software College, Northeastern University, Shenyang 110819, China

20223246@stu.neu.edu.cn, dulun.dl@antgroup.com
{20226504,2401925}@stu.neu.edu.cn, sunhz6@mails.neu.edu.cn
heli@cs.aau.dk, {yuminghe,tczhang,yuge}@mail.neu.edu.cn

## ABSTRACT

Interactive theorem provers (ITPs) require manual formalization, which is labor-intensive and demands expert knowledge. While automated formalization offers a potential solution, it faces two major challenges: *model hallucination* (e.g., undefined predicates, symbol misuse, and version incompatibility) and the *semantic gap* caused by ambiguous or missing premises in natural language descriptions. To address these issues, we propose **CRAMF**, a Concept-driven Retrieval-Augmented Mathematical Formalization framework. CRAMF enhances LLM-based autoformalization by retrieving formal definitions of core mathematical concepts, providing contextual grounding during code generation. However, applying retrieval-augmented generation (RAG) in this setting is non-trivial due to the lack of structured knowledge bases, the polymorphic nature of mathematical concepts, and the high precision required in formal retrieval. We introduce a framework for automatically constructing a concept-definition knowledge base from Mathlib4, the standard mathematical library for the Lean 4 theorem prover, indexing over 26,000 formal definitions and 1,000+ core mathematical concepts. To address conceptual polymorphism, we propose contextual query augmentation with domain- and application-level signals. In addition, we design a dual-channel hybrid retrieval strategy with reranking to ensure accurate and relevant definition retrieval. Experiments on miniF2F, ProofNet, and our newly proposed AdvancedMath benchmark show that CRAMF can be seamlessly integrated into LLM-based autoformalizers, yielding consistent improvements in translation accuracy—achieving up to 62.1% and an average of 29.9% relative improvement.

## 1 INTRODUCTION

Automated formalization is the process of translating natural language descriptions of mathematical theorems into formally verifiable representations (Weng et al., 2025), such as Lean (Moura & Ullrich, 2021), Coq (Huet et al., 1997), or Isabelle (Paulson, 1994). In the era of large language models (LLMs), it serves as a crucial bridge between informal human reasoning and formal symbolic logic, enabling AI systems to participate meaningfully in mathematical problem solving. Its importance is exemplified by DeepMind's AlphaProof, which achieved silver-medal-level performance in the 2024 International Mathematical Olympiad by leveraging an end-to-end formalization pipeline based on the Lean theorem prover (AlphaProof & AlphaGeometry, 2024). As LLMs become central to automated theorem proving, the accuracy and reliability of automated formalization directly impact the overall success of proof generation.

---

[*]These authors contributed to the work equllly and should be regarded as co-first authors.
[†]Corresponding author.

Current mainstream approaches to automated formalization rely on Large Language Models (LLMs) to directly translate natural language into formal mathematical statements (Wu et al., 2022). Typical strategies include few-shot prompting of pre-trained models and fine-tuning on aligned natural language–formal language (NL–FL) pairs (Xin et al., 2024). While recent systems, such as Herald (Gao et al., 2024b) and Kimina-Prover (Wang et al., 2025), have shown promising results, they continue to face two fundamental challenges. 1) **Model hallucination** arises when LLMs generate confident but incorrect formal code. Common failure modes include fabricating undefined concepts in Mathlib (Lean's standard library), misusing symbols due to informal reasoning, and producing outdated definitions incompatible with the latest Mathlib version. 2) **Semantic gap** stems from the mismatch between the ambiguity of natural language and the precision of formal languages. A key difficulty is *conceptual polymorphism*, where identical expressions correspond to different formal definitions depending on context, domain, or abstraction level. This often leads to inaccurate formalizations, especially in applied fields like combinatorics, where essential entities are frequently implicit and hard to recover from surface text. **Figure 1** provides concrete examples of these dual challenges.

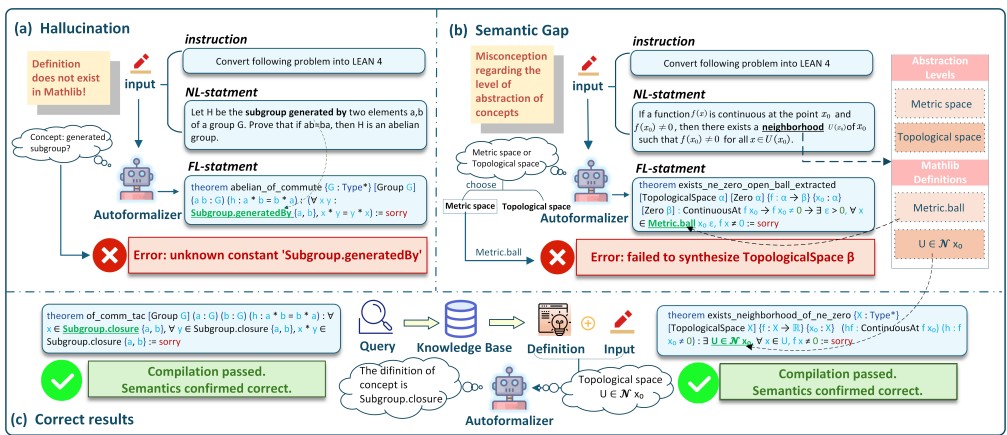

Figure 1: Examples illustrating model hallucination and semantic gap. Panel (a) demonstrates a case of autoformalization failure due to the model fabricating an undefined predicate in Mathlib, resulting in compilation errors. Panel (b) exhibits a semantic gap arising from the conceptual polymorphism of "neighborhood" (defined differently in topological versus metric spaces) where the model fails to recognize the appropriate abstraction level, triggering type class synthesis errors during compilation. Panel (c) showcases correct formalization results generated by the CRAMF framework, which effectively resolves both issues through structured knowledge grounding and context-aware concept resolution.

In general-domain natural language processing, similar challenges, such as factual hallucination and context-sensitive ambiguity, are often addressed through Retrieval-Augmented Generation (RAG) (Gao et al., 2023b). By retrieving relevant external knowledge to ground and guide model outputs, RAG has proven effective in improving factual consistency and semantic precision across a range of tasks. Despite this success, the application of RAG to mathematical automated formalization remains largely unexplored. This work investigates whether retrieval-based methods can be adapted to improve the reliability and accuracy of LLM-driven formalization. However, directly applying RAG in this domain introduces new obstacles. First, unlike encyclopedic knowledge in the general domain, mathematical libraries such as Mathlib lack structured, queryable mappings from natural language expressions to formal definitions, making effective retrieval non-trivial. Second, resolving *conceptual polymorphism* requires not just retrieving related content, but disambiguating between multiple context-sensitive formalizations, a task that standard RAG pipelines are not equipped to handle. These limitations call for domain-specific retrieval augmentation tailored to the needs of formal reasoning systems.

To systematically address hallucination and semantic gap in Lean-based autoformalization, we propose the **Concept-driven Retrieval-Augmented Mathematical Formalization (CRAMF)** framework. CRAMF enhances formalization accuracy by retrieving precise definitions of core mathematical concepts from Mathlib to provide contextual grounding for LLM-based autoformalizers. At

its core, CRAMF relies on a structured knowledge base that explicitly maps natural language expressions to their corresponding formal definitions. We define a schema for this concept-definition knowledge base that captures the many-to-many relationships between informal descriptions and formal representations. To support scalability and coverage, we design an automated pipeline that constructs the knowledge base by aligning Mathlib definitions with diverse, canonical natural language expressions. To address conceptual polymorphism, CRAMF augments user queries with contextual signals and leverages a hybrid retrieval strategy to improve definition disambiguation. By incorporating domain-specific cues and application context, the system enhances its ability to distinguish between multiple candidate definitions of the same concept. A combination of symbolic and semantic retrieval, followed by reranking, ensures accurate and context-aware retrieval of formal definitions.

In conclusion, our contributions are as follows:

- We propose the **Concept-driven Retrieval-Augmented Mathematical Formalization (CRAMF)** framework, which retrieves precise formal definitions of core mathematical concepts to provide contextual grounding for LLM-based autoformalization.

- We define a structured concept-definition knowledge base covering over 26,000 Mathlib definitions and 1,000+ core mathematical concepts, and develop an automated LLM-powered pipeline to construct this resource by aligning formal definitions with diverse natural language expressions.

- We demonstrate that CRAMF serves as a plug-and-play enhancement for LLM-based autoformalizers, consistently improving translation accuracy on miniF2F (Zheng et al., 2021), ProofNet (Azerbayev et al., 2023), and our proposed AdvancedMath benchmark—achieving up to 62.1% and an average of 29.9% relative improvement.

## 2 METHOD

This section presents the proposed **Retrieval-Augmented Mathematical Formalization Framework (CRAMF)**. CRAMF operates under a retrieval-augmented paradigm: given a natural language description of a mathematical theorem, it first retrieves formal definitions of relevant concepts from a structured knowledge base, then supplies them as contextual grounding to an LLM-based autoformalizer. This enriched context helps the model generate Lean 4 code that is syntactically correct, semantically aligned with the theorem, and compliant with Mathlib specifications. We describe the framework through three main components: concept-definition knowledge base construction, mathematical concept extraction(MCE), and definition retrieval, which comprises the Dual-Pathway Hybrid Retrieval (DHR) module and a reranking module. An overview of CRAMF is illustrated in Figure 2.

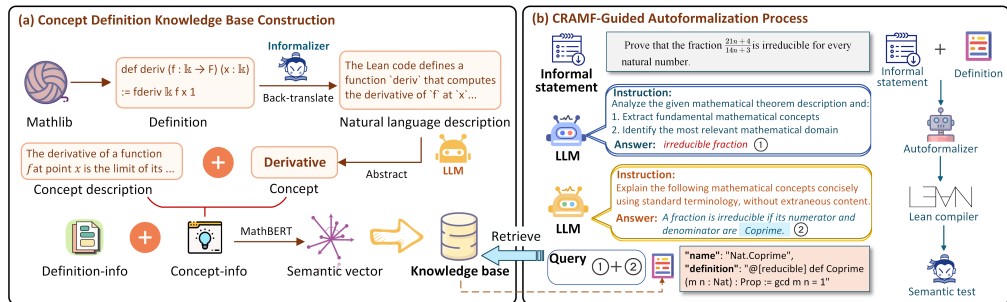

Figure 2: Overview of the CRAMF framework. (a) Construction of the concept-definition knowledge base via back-translation and concept extraction from Mathlib definitions. (b) Integration into autoformalization: extracted concepts retrieve relevant definitions to guide formal code generation.

## 2.1 Concept Definition Knowledge Base Construction

### 2.1.1 Ontology Schema Design

To support structured representation and efficient retrieval of Lean-based mathematical knowledge, we define a lightweight ontology $\mathcal{O}$ and implement a corresponding relational database schema for storage and querying. The ontology is formalized as a quadruple:

$$\mathcal{O} = (\mathcal{C}, \mathcal{P}, \mathcal{A}, \mathcal{R})$$

where $\mathcal{C}$ denotes the set of entity types, $\mathcal{P}$ is the set of attributes, $\mathcal{A} : \mathcal{C} \to \mathbb{P}(\mathcal{P})$ is the attribute mapping function that assigns to each entity type its associated attributes, and $\mathcal{R}$ is the set of semantic relations. Here, $\mathbb{P}(\mathcal{P})$ represents the power set of $\mathcal{P}$.

To explicitly capture concept-definition mappings and support retrieval, we define the entity types as:

$$\mathcal{C} = \{\Gamma, \Theta, \Phi\},$$

where $\Gamma$ represents abstract mathematical concepts, $\Theta$ denotes formal mathematical definitions extracted from Mathlib, and $\Phi$ consists of natural language annotations associated with those definitions. Each entity type is characterized by a set of attributes via the mapping function $\mathcal{A}$, which serve as the logical basis for database fields: $\Gamma$ is characterized by the attribute triple $\langle \gamma_n, \gamma_d, \gamma_e \rangle$, representing the concept's name, domain, and explanatory description; $\Theta$ is characterized by $\langle \theta_\tau, \theta_f, \theta_p \rangle$, denoting the definition's identifier, formal expression, and module path; $\Phi$ is defined by the singular attribute $\phi_a$ representing definition's natural language annotation.

The relation set $\mathcal{R}$ defines the semantic links between entities:

$$\mathcal{R}_1 \subseteq \Gamma \times \mathbb{P}(\Phi),$$

$$\mathcal{R}_2 : \Phi \to \Theta,$$

where $\mathcal{R}_1$ establishes a one-to-many relationship between entity $\Gamma$ and entity $\Theta$, signifying that a single mathematical concept may correspond to multiple Lean definitions. $\mathcal{R}_2$ defines a one-to-one mapping between entity $\Theta$ and entity $\Phi$, denoting that each formal definition corresponds to exactly one unique natural language explanation.

### 2.1.2 Knowledge Base Population

We automate the construction of the knowledge base by instantiating the ontology schema with specific record instances extracted from Mathlib. Guided by the attribute mapping function $\mathcal{A}$, we populate the three entity types $\mathcal{C} = \{\Gamma, \Theta, \Phi\}$ and establish semantic relations $\mathcal{R}$ among them. We begin by parsing Mathlib using Lean 4's official documentation tool, `doc-gen4`, to extract all definitions declared via `def`, `class`, or `structure`. These populate the attributes of the *Definition* $\theta$ and *Description* $\Phi$ entities, including identifiers, formal representations, module paths, and associated annotations.

To instantiate the mathematical *Concept* entity $\Gamma$, we employ a reverse translation strategy using the pre-trained language model InternLM-Math-7B (Ying et al., 2024b). Each formal definition is passed to the model to generate natural language descriptions. We apply a self-consistency validation procedure to improve generation quality, producing three candidate descriptions and selecting the one most semantically aligned with the original annotation. Subsequently, the selected natural language description is processed by a concept extraction model (DeepSeek-V3) to identify the underlying mathematical concept and generate an explanatory gloss. This final step completes the construction of the $\Gamma$ entity. Illustrative examples of this end-to-end process are shown in Figure 3.

### 2.1.3 Vector Encoding and Index Construction

Considering that the MathBERT (Peng et al., 2021) model is pre-trained based on large-scale mathematical corpus, which helps to accurately capture the semantics of mathematical concepts, We employ it to perform semantic encoding on the $\gamma_e$ field of $\Gamma$ and the $\theta_a$ field of the descriptions in the knowledge base as follows:

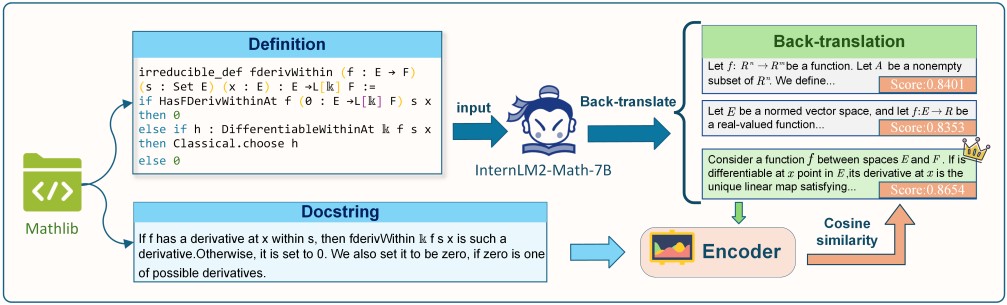

Figure 3: The example illustrates reverse translation of Lean's defintion of *fderivWithin* via InternLM-plus7B, with semantic similarity scoring against Mathlib annotations.

$$v_c = MathBERT(\gamma_e)$$
$$v_d = MathBERT(\theta_a)$$

An efficient vector index is then built using Faiss (Douze et al., 2024). Ultimately, each knowledge unit is represented as the following quadruple:

$$(\gamma_n, \theta_r, v_c, v_d)$$

where $\gamma_n$ is the core mathematical concept; $\theta_r$ is its corresponding Mathlib definition name; $v_c$ and $v_d$ are the respective semantic vectors.

## 2.2 CONCEPT EXTRACTION

In CRAMF, mathematical concept extraction(MCE) serves as the bridge between natural language problem descriptions and formal definitions. The objective is to accurately identify the core mathematical concepts from the input natural language theorem description, providing fundamental query units for subsequent definition retrieval.

Mathematical problems vary in explicitness: some contain explicit concept mentions, while others rely on implicit mathematical structures. We therefore employ distinct extraction strategies.

**For conventional proof problems explicitly containing mathematical concepts**: We leverage the powerful comprehension capabilities of large language models (LLMs) to directly identify and extract the core concepts within the mathematical theorem. **For applied mathematics problems with implicit mathematical concepts**: Their problem descriptions typically lack explicit mathematical concept keywords. For example, the combinatorial problem "Prove: Among any 6 people, there are always at least 3 people who either all know each other or are all strangers to each other" implicitly involves the mathematical concept of a graph. When handling such problems, we introduce an LLM-based problem rewriting mechanism. During the concept extraction phase, this mechanism performs mathematical modeling and rewriting of the original problem to explicitly express its core mathematical structure before concept extraction proceeds. Details on prompt construction and examples are provided in the appendix.

## 2.3 DEFINITION RETRIEVAL

Due to the conceptual polymorphism of mathematical concepts, the interpretation of the same naturally described concept can vary across different abstraction levels and semantic granularities. Without providing relevant context, LLMs struggle to confirm the precise formal definition corresponding to a mathematical concept when generating formal statements. To address this, we introduce a two-stage retrieval process comprising query enhancement and dual-pathway hybrid retrieval.

### 2.3.1 QUERY ENHANCEMENT

When a concept has multiple formal definitions, using solely the concept as the query can lead to excessive noise and low recall. We therefore enhance queries via conceptual parsing. Using LLMs

under strict prompting, we perform conceptual parsing of the mathematical theorem, generating an interpretation of terms based on the extracted core concepts and the existing theorem description. The concept is then concatenated with this terminological interpretation to form the query. The model-generated interpretation incorporates domain information and application context for the concept, while also implying dependencies between concepts. Using this as the query enables matching richer semantics during retrieval, helping to mitigate inaccuracies caused by conceptual polymorphism and bridging the semantic gap between natural language descriptions and Mathlib's specialized definitions.

### 2.3.2 DUAL-PATHWAY HYBRID RETRIEVAL

We adopt a collaborative strategy combining hybrid retrieval and reranking, integrating semantic vector retrieval with exact matching mechanisms to ensure retrieval results satisfy both semantic relevance and formal precision. Specifically, the system executes the following two retrieval pathways in parallel:

- **Symbol-Level Keyword Matching**: The LLM generates search keywords for the extracted core mathematical concept, which are used for exact matching against Mathlib definition symbols via regular expressions, forming a base candidate set.

- **Semantic Similarity Retrieval**: The query text, composed of the concept and its interpretation, is input into a MathBERT encoder to calculate its vector similarity with concept explanations in the knowledge base, initially recalling the Top-10 similar concepts. These are then reranked using the bge-reranker-v2-m3 (Guo et al., 2024) model to filter the semantically most relevant Top-5 concepts. The definitions corresponding to these concepts are merged into the base candidate set.

To further optimize prompt quality, a reranking mechanism is introduced. The bge-reranker-v2-m3 (Guo et al., 2024) model performs fine-grained semantic assessment of candidate definitions. By calculating the similarity between the conceptual interpretation within the query and the annotations of candidate definitions, it reranks the candidates. The Top-3 definitions with the highest semantic match are ultimately selected to constitute the context prompt for the automated formalization task.

## 3 EXPERIMENTS

In this section, we conduct experiments to address the following research questions: **RQ1:** How effectively does CRAMF framework improve compilation success rate and formalization accuracy in autoformalizing natural language mathematical theorems to Lean 4? **RQ2:** How does CRAMF framework perform in definition retrieval compared to baseline retrieval methods? **RQ3:** What are the individual contributions of CRAMF's core components to the final formalization performance?

### 3.1 EXPERIMENTAL SETUP

#### 3.1.1 DATASETS

We evaluate our method on two public datasets (miniF2F (Zheng et al., 2021) and ProofNet (Azerbayev et al., 2023)) and one proprietary dataset (AdvancedMath). AdvancedMath comprises 173 informal proof problems in higher mathematics, selected from the authoritative textbook "Advanced Mathematics (8th Edition)" published by Tongji University. It features standardized problem formulations and covers core areas such as spatial analytic geometry, vector algebra, series theory, and ordinary differential equations. In comparison to datasets like miniF2F and ProofNet, AdvancedMath presents a moderate level of difficulty and integrates logical reasoning, symbolic computation, and natural language understanding, making it a suitable benchmark for evaluating the comprehensive capabilities of formal translation systems. We release this dataset publicly to support future research in this area.[*]

---

[*]The dataset is available at: `https://github.com/kahvia0526/CRAMF`

### 3.1.2 BASELINE

**Autoformalization Models** We evaluate the effectiveness of the CRAMF framework against several baseline autoformalization models: the open-source autoformalizers Herald-7B (Gao et al., 2024b), Kimina-7B (Wang et al., 2025) and Godel-Formalizer-V2-8B (Lin et al., 2025), as well as the powerful API-based LLMs DeepSeek-V3 (DeepSeek et al., 2024), GPT-4o, Gemini 2.5 Flash, and Claude 4.1.

**Retrieval Models** We evaluate several retrieval methods, including the standard RAG baselines BM25 (Robertson et al., 2009), Rewrite-Retrieve-Read (Ma et al., 2023), and HyDE (Gao et al., 2023a), as well as the LeanSearch (Gao et al., 2024a) and LeanExplore (Asher, 2025) engines, which provide semantic search for Lean 4 declarations and mathlib4. We also include Dependency Retrieval (DR) (Liu et al., 2025), an open-source autoformalization-oriented model that employs a dependency-based retrieval mechanism, to comprehensively compare definition retrieval capabilities across multiple datasets.

### 3.1.3 EVALUATION METHODS

**Autoformalization Evaluation** Given the prohibitive cost and limited scalability of manual evaluation for large-scale experiments, we selected the evaluation pipeline from the LeanWorkBook project (Ying et al., 2024a), whereas manual assessment (Sidhu et al., 2024), despite offering deeper qualitative insights, was deemed impractical. Formalized outputs are verified by the Lean compiler. Results that compile successfully are then back-translated into natural language descriptions using the InternLM2-Math-Plus-7B model (Ying et al., 2024b). Finally, DeepSeek-V3 assesses the semantic consistency between the back-translated statements and the original informal statements.

**Retrieval Performance Evaluation** We evaluate retrieval quality using two metrics: Average Contribution Score (ACS) and Relevant Definition Hit Rate (HitRate@K).

**1) Average Contribution Score (ACS)** measures the overall relevance of retrieved definitions to the target problem. Each retrieved definition is assigned a score from 0 to 3 based on its contribution to the formalization process. A score of **3 (Exact Match)** indicates that the definition appears in the final formalized code, compilation succeeds, semantic consistency assessment passes, and the generated Lean 4 expression closely aligns with the original problem. A score of **2 (Strong Relevance)** is given when the definition is not used in the final code but is semantically or mathematically related to the problem statement, as judged by a large language model. A score of **1 (Weak Relevance)** is assigned when the definition shares topical relevance but lacks direct semantic connection (also evaluated by an LLM). A score of **0 (Erroneous Reference)** indicates that the definition appears in the code but either fails to compile or fails the semantic consistency assessment.

Scores of 3 and 0 are assigned automatically using a combination of regular expression matching and the outcomes of compilation and semantic consistency verification. Scores of 2 and 1 are assessed using DeepSeek-R1; see the Appendix for prompts. Given a sample set $T = \{t_1, t_2, ..., t_n\}$, where each problem $t_i$ has a set of retrieved definitions $\mathcal{D}_i = \{d_{i1}, d_{i2}, ..., d_{ik}\}$, the ACS is defined as:

$$ACS = \frac{1}{\sum_{i=1}^{n} |\mathcal{D}_i|} \sum_{i=1}^{n} \sum_{j=1}^{|\mathcal{D}_i|} \text{Score}(d_{ij}, t_i) \tag{1}$$

where $\text{Score}(d_{ij}, t_i) \in \{0, 1, 2, 3\}$ denotes the contribution level of definition $d_{ij}$ to problem $t_i$.

**2) Relevant Definition Hit Rate (HitRate@K)** measures whether at least one high-quality definition appears among the Top-$K$ retrieved results for each problem. Specifically, we check whether any of the Top-$K$ definitions achieve a score of 3. Formally, this metric is defined as:

$$HitRate@K = \frac{1}{n} \sum_{i=1}^{n} \mathcal{I} \left[ \max_{j=1,...,K} \text{Score}(d_{ij}, t_i) = 3 \right] \tag{2}$$

where $n$ is the total number of problems and $\mathcal{I}[\cdot]$ is the indicator function (1 if the condition holds, 0 otherwise). In our experiments, we set $K = 3$. This metric captures the proportion of problems for which the Top-$K$ retrieved definitions contain at least one that is strongly relevant or an exact match.

Table 1: Compilation Pass Rate@10 of each base model (Orig.) versus the same model augmented with CRAMF on three datasets. **RG** (Relative Gain) represents the relative improvement rate.

| Model | MiniF2F | | | ProofNet | | | AdvancedMath | | |
|---|---|---|---|---|---|---|---|---|---|
| | Base | +CRAMF | RG | Base | +CRAMF | RG | Base | +CRAMF | RG |
| Deepseek-V3 | 52.3% | 69.3% | +32.5% | 39.8% | 53.1% | +33.4% | 31.7% | 48.2% | **+52.1%** |
| GPT-4o | 70.1% | 82.7% | +18.0% | 48.0% | 62.6% | +30.4% | 48.5% | 59.1% | +21.9% |
| Gemini 2.5 Flash | 82.4% | 89.3% | +8.4% | 70.9% | 75.1% | +5.9% | 79.3% | 82.1% | +3.5% |
| Claude 4.1 | 95.1% | 98.0% | +3.0% | 81.1% | 85.6% | +5.5% | 98.8% | 99.4% | +0.6% |
| Herald-7B | 79.1% | 93.6% | +18.3% | 60.9% | 79.4% | +30.4% | 80.0% | 90.2% | +12.8% |
| Kimina-7B | 98.8% | 99.2% | +0.4% | 87.4% | 94.1% | +7.7% | 98.8% | 100% | +1.2% |
| Godel-V2-8B | 99.6% | 100.0% | +0.4% | 85.6% | 87.4% | +2.1% | 98.8% | 100% | +1.2% |

Table 2: Formalization Accuracy Rate@10 of each base model (Orig.) versus the same model augmented with CRAMF on three datasets. **RG** (Relative Gain) represents the relative improvement rate.

| Model | MiniF2F | | | ProofNet | | | AdvancedMath | | |
|---|---|---|---|---|---|---|---|---|---|
| | Base | +CRAMF | RG | Base | +CRAMF | RG | Base | +CRAMF | RG |
| Deepseek-V3 | 36.9% | 47.1% | +27.6% | 23.6% | 37.0% | +56.8% | 19.8% | 32.1% | **+62.1%** |
| GPT-4o | 49.6% | 60.1% | +21.2% | 29.9% | 42.2% | +41.1% | 22.8% | 34.7% | +52.2% |
| Gemini 2.5 Flash | 72.1% | 82.0% | +13.7% | 57.0% | 63.1% | +10.7% | 43.9% | 50.9% | +15.9% |
| Claude 4.1 | 83.6% | 92.6% | +10.8% | 72.2% | 79.7% | +10.4% | 57.2% | 65.9% | +15.2% |
| Herald-7B | 49.2% | 63.1% | +28.3% | 44.4% | 55.1% | +24.1% | 39.3% | 51.4% | +30.8% |
| Kimina-7B | 80.3% | 84.8% | +5.6% | 65.0% | 69.3% | +6.6% | 61.3% | 66.5% | +8.5% |
| Godel-V2-8B | 88.1% | 95.1% | +7.9% | 73.0% | 80.2% | +9.9% | 57.8% | 68.2% | +18.0% |

## 3.2 AUTOFORMALIZATION PERFORMANCE (RQ1)

Our evaluation assesses CRAMF's impact on autoformalization models through two metrics: Compilation Pass Rate@10 (CPR@10) and Formalization Accuracy Rate@10 (FAR@10). Comparative results (Table 1 and Table 2) demonstrate that CRAMF universally enhances model performance, though the extent of improvement is architecture-dependent.

Particularly notable are the substantial gains observed in the general large model DeepSeek-V3 without domain-specific fine-tuning, indicating that CRAMF effectively fills formalization knowledge gaps in general-purpose models through precise definition retrieval, constructing a "plug-and-play" formal knowledge bridge that substantially lowers the domain entry barrier. On the AdvancedMath dataset involving multiple complex concepts, the relative improvement rates for both compilation pass rate and formalization accuracy reach their highest values at 52.1% and 62.1% respectively. The significant improvement in compilation pass rate confirms that injecting core mathematical concept definitions enables large models to express formal symbols with greater precision, while the enhancement in formalization accuracy reflects deeper semantic comprehension of mathematical problems.

## 3.3 KNOWLEDGE BASE AND RETRIEVAL PERFORMANCE (RQ2)

To assess the efficacy of the concept-definition knowledge base and the hybrid retrieval strategy in CRAMF, we evaluate its performance in accurately retrieving core mathematical definitions for theorem autoformalization.

Table 3 and Table 4 report the average contribution score (ACS) and the Top-3 hit rate (HitRate@3) of different retrieval frameworks across three datasets. As shown in both tables, CRAMF significantly outperforms all baselines, achieving an average improvement of more than 0.5 points in contribution score and up to a 15-percentage-point increase in HitRate@3 over the best-performing baselines. Notably, CRAMF maintains robust performance on the AdvancedMath dataset, underscoring its strong generalization capability in complex mathematical contexts.

Table 3: Average contribution scores (ACS) of retrieval methods on three datasets.

| Method | MiniF2F | ProofNet | AdvancedMath |
|---|---|---|---|
| BM25 | 0.91 | 0.94 | 0.83 |
| R3 | 1.38 | 1.45 | 1.31 |
| HyDE | 1.55 | 1.61 | 1.52 |
| LeanExplore | 1.42 | 1.51 | 1.35 |
| LeanSearch | 1.58 | 1.63 | 1.56 |
| DR | 1.39 | 1.47 | 1.26 |
| CRAMF | **2.07** | **2.14** | **1.93** |

Table 4: Top-3 hit rate of relevant definitions (HitRate@3) for each retrieval framework across three datasets.

| Method | MiniF2F | ProofNet | AdvancedMath |
|---|---|---|---|
| BM25 | 11.3% | 14.5% | 9.1% |
| R3 | 19.6% | 21.1% | 15.4% |
| HyDE | 32.4% | 35.5% | 30.7% |
| LeanExplore | 31.5% | 37.7% | 23.8% |
| LeanSearch | 36.8% | 42.3% | 35.1% |
| DR | 32.6% | 33.8% | 28.5% |
| CRAMF | **44.2%** | **50.6%** | **42.9%** |

Table 5: Ablation results of CRAMF submodules evaluated by FAR@10.

| Method | MiniF2F | ProofNet | AdvancedMath |
|---|---|---|---|
| CRAMF | 63.1% | 55.1% | 51.4% |
| w/o MCE | 58.2% | 49.8% | 44.0% |
| w/o DHR | 48.5% | 36.8% | 37.1% |
| w/o Rerank | 54.3% | 47.5% | 43.9% |

Table 6: Effect of structured problem rewriting on FAR@10 over 241 combinatorics problems in CombMath.

| Method | CombMath |
|---|---|
| CRAMF | 63.1% |
| w/o ReWrite | 54.3% |

In contrast, BM25 relies on surface-level keyword matching, making it difficult to distinguish between the semantic variations of the same concept across different levels of abstraction. The Rewrite-Retrieve-Read approach improves upon BM25 by enhancing recall of implicitly mentioned concepts through query rewriting, yet it still falls short of satisfying the symbolic precision required by Lean. While HyDE achieves better ACS than the preceding baselines, it is prone to hallucination, often retrieving incorrect definitions that ultimately limit its practical utility. Although LeanSearch and LeanExplore serve as semantic search engines tailored for the Lean environment, their retrieval results primarily return theorems rather than definitions. While they achieve higher HitRate@3 than traditional RAG methods, they still exhibit limitations in handling cross-domain and polymorphic concepts. On the other hand, Dependency Retrieval (DR) leverages dependency associations among formal objects in the Mathlib for retrieval. Although this design can capture topological connections between definitions, it is inherently constrained by surface-level syntactic structures. The dependency graph struggles to model the deep semantic correspondence between natural language and formal concepts, resulting in suboptimal retrieval performance despite being explicitly trained for autoformalization. These limitations highlight the advantages of CRAMF's structured knowledge base and hybrid retrieval mechanism in ensuring both semantic relevance and formal accuracy.

## 3.4 ABLATION STUDY (RQ3)

To dissect the individual contributions of key submodules within the CRAMF framework, particularly considering the condition-triggered problem rewriting mechanism in our retrieval pipeline, we conduct a structured ablation study across two experimental settings. 1) On three general datasets, we evaluate the impact of three core components: Mathematical Concept Extraction (MCE), Dual-Pathway Hybrid Retrieval(DHR), and reranking module. 2) We assess the effectiveness of the rewriting mechanism in handling problems characterized by high representational abstraction and implicit terminology using CombMath.[†] Using Herald-7B (Gao et al., 2024b) as the backbone generation model, We summarize the ablation results on the general datasets in Table 5. The removal of the DHR module leads to a substantial decline of 15.7% in average FAR, underscoring the necessity of simultaneously capturing semantic relevance and symbolic precision in mathematical formal environments. The MCE module also proves indispensable, with its removal causing a 6-7 percentage point reduction in FAR, demonstrating that providing domain knowledge and application contexts for mathematical concepts effectively resolves definition ambiguity. Furthermore, as shown in Ta-

---

[†]The dataset of 241 combinatorics problems is derived from Combinatorics: The Art of Counting (Sagan, 2020), an authoritative source covering core subdomains like enumerative combinatorics and combinatorial design.

ble 6, activating the rewriting module yields an 8.8% improvement in FAR, validating the utility of our conditionally-triggered rewriting strategy for semantic modeling in mathematically complex scenarios.

## 4 RELATED WORKS

### 4.1 AUTOFORMALIZATION

Autoformalization refers to the process of transforming informal mathematical statements into formally verifiable representations (Jiang et al., 2023; Poiroux et al., 2024; Wang et al., 2020; Tian et al., 2025; Li et al., 2025). Existing approaches can be broadly categorized into rule-based and LLM-based methods. Rule-based methods leverage explicit logical and syntactic rules (Weng et al., 2025), often adopting controlled natural languages (e.g., Mizar (Rudnicki, 1992), ForTheL (Vershinin & Paskevich, 2000)) or grammatical frameworks (e.g., GF (Pathak, 2024)) to construct abstract syntax trees that are then translated into formal code. In contrast, LLM-based approaches utilize few-shot prompting (Wu et al., 2022) or fine-tuning on aligned natural language–formal language (NL–FL) pairs (Xuejun et al., 2025).

### 4.2 RETRIEVAL-AUGMENTED GENERATION

Retrieval-Augmented Generation (RAG) enhances LLM outputs by integrating external knowledge retrieved in response to the input query, thereby reducing hallucinations and factual errors (Li et al., 2024). In the context of formal reasoning, RAG has been applied to mathematical autoformalization and verification tasks. Azerbayev et al. (2023) propose a retrieval-augmented formalization method that selects prompt-relevant statements to improve proposition generation. RAutoformalizer Liu et al. (2025) adopts retrieval-based augmentation to support statement-level formalization. Similarly, ReProver (Yang et al., 2023) incorporates a premise selection module that filters relevant auxiliary statements from large formal corpora to aid theorem generation. In a closely related vein, Tao et al. (2025) introduce a lightweight premise retrieval model trained on Mathlib4, which employs a tokenizer specifically designed for formal language and a fine-grained similarity measure to enhance retrieval accuracy in Lean-based theorem proving.

## 5 CONCLUSION

This paper proposes CRAMF, a concept-driven retrieval-augmented framework for mathematical formalization. By leveraging a structured knowledge base from Mathlib4 and a tailored retrieval pipeline, CRAMF effectively mitigates model hallucinations and semantic inprecision, significantly improving formalization accuracy across diverse benchmarks. Future work may extend to more complex mathematical domains and explore cross-library, multi-step reasoning, and feedback-incorporated retrieval augmentation strategies.

## 6 REPRODUCIBILITY STATEMENT

To ensure the reproducibility of our work, we have included the complete source code in the supplementary materials. An anonymized version of the code is also available at the following link: `https://github.com/kahvia0526/CRAMF`. The detailed experimental setup, hyperparameters, and prompts used for the large language models are provided in the appendix.

### ACKNOWLEDGMENTS

This research was funded by the National Natural Science Foundation of China (Nos. 62272093, 62137001)

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

## A    ADDITIONAL EXPERIMENTAL RESULTS

### A.1    STATISTICAL SIGNIFICANCE TESTS

To verify whether the performance gains brought by the CRAMF framework are statistically significant across different models and datasets, we conduct the Wilcoxon signed-rank test to compare each model's performance with and without CRAMF.

The Wilcoxon signed-rank test is a non-parametric statistical method used for analyzing differences between two paired samples. It is particularly suitable for our setting: evaluating the same model under two conditions (Base vs. +CRAMF) on a per-problem basis.

We encode the performance on each sample (mathematical problem) as a binary outcome: if at least one of the Top-10 generated results passes the semantic consistency check, the sample is considered successful (assigned 1); otherwise, it is considered unsuccessful (assigned 0). These paired binary results are then used to assess the performance difference.

We use FAR@10 as the evaluation metric. Each question in a dataset constitutes one sample, and each sample has two associated binary values (Base vs. Base+CRAMF). The statistical test proceeds as follows:

1. Compute the difference for each paired sample.

2. Discard pairs with zero difference; rank the remaining pairs by absolute difference.

3. Calculate the sum of ranks for positive and negative differences; the Wilcoxon test statistic is the smaller of the two.

4. Given a significance level of $\alpha = 0.01$, compute the corresponding p-value.

### A.1.1    RESULTS

As shown in Table 7, all $p$-values are below the significance threshold $\alpha = 0.01$, indicating that the observed improvements are statistically significant across all evaluated settings. The CRAMF framework achieves statistically significant improvements on most model-dataset combinations. General-purpose models such as DeepSeek-V3 and GPT-4o benefit the most, indicating that CRAMF effectively supplements their lack of mathematical domain knowledge. Even for high-performing customized models like Kimina-7B, CRAMF still yields lightweight but significant gains. These results demonstrate that CRAMF provides systematic enhancement to mathematical formalization, rather than accidental variation.

### A.2    CASE STUDY

This subsection demonstrates the efficacy of CRAMF in improving automated formalization accuracy through comparative analysis of baseline failure cases versus CRAMF-augmented success cases.

#### A.2.1    EXAMPLES FROM THE COMBINED DATASET THAT REQUIRE TRIGGERING THE REWRITING MECHANISM

This part demonstrates the operational workflow of the rewriting mechanism when handling mathematical problems lacking explicitly stated concepts.

Table 7: Wilcoxon signed-rank test results comparing Base and +CRAMF across models and datasets. Each row corresponds to one model-dataset pair. The **Base (%)** and **+CRAMF (%)** columns show the FAR@10 performance without and with CRAMF, respectively. $n$ indicates the number of samples (math problems) in the dataset. The **Wilcoxon** column gives the signed-rank test statistic (smaller of the rank sums), and the $p$-**value** column reports the corresponding $p$-value. All $p$-values are below the significance threshold $\alpha = 0.01$, confirming that the observed gains are statistically significant.

| Model | Dataset | $n$ | Base (%) | +CRAMF (%) | Wilcoxon | $p$-value |
|---|---|---|---|---|---|---|
| DeepSeek-V3 | miniF2F | 244 | 36.9 | 47.1 | 78 | 2.7e-04 |
| DeepSeek-V3 | ProofNet | 373 | 23.6 | 37.0 | 1834 | 2.0e-08 |
| DeepSeek-V3 | AdvancedMath | 173 | 19.8 | 32.1 | 586 | 8.1e-06 |
| GPT-4o | miniF2F | 244 | 49.6 | 60.1 | 1121 | 6.3e-04 |
| GPT-4o | ProofNet | 373 | 29.9 | 42.2 | 1905 | 1.7e-07 |
| GPT-4o | AdvancedMath | 173 | 22.8 | 34.7 | 524 | 4.1e-05 |
| Herald-7B | miniF2F | 244 | 49.2 | 63.1 | 595 | 3.0e-09 |
| Herald-7B | ProofNet | 373 | 44.4 | 55.1 | 820 | 1.0e-10 |
| Herald-7B | AdvancedMath | 173 | 39.3 | 51.4 | 231 | 2.0e-06 |
| Kimina-7B | miniF2F | 244 | 80.3 | 84.8 | 66 | 4.5e-04 |
| Kimina-7B | ProofNet | 373 | 65.0 | 69.3 | 106 | 1.6e-04 |
| Kimina-7B | AdvancedMath | 173 | 61.3 | 66.5 | 45 | 1.3e-03 |

**The Case for Rewrite mechanism**

NL-statement:

Prove that among any 10 people, if there do not exist 3 people who are all mutually unacquainted, then there must exist 4 people who all know each other. Likewise, if there do not exist 3 people who are all mutually acquainted, then there must exist 4 people who are all mutually unacquainted.

Rewrite:

Let be a complete graph where each edge is colored red (acquaintance) or blue (non-acquaintance). Prove that:
If there is no blue triangle in the graph, then there must exist a red
If there is no red triangle, then there must exist a blue .

Abstract concept: Complete Graph, Edge-coloring, Ramsey theory, Monochromatic Subgraph, monochromatic triangle

Figure 4: Example from the rewriting mechanism

Figure 4 demonstrates the need for the rewriting mechanism when mathematical concepts are implicit. The original informal statement lacks formal terms like "grap" or "Ramsey theory". Rewriting models it as a complete edge-colored graph, enabling the extraction of key abstract concepts and bridging informal language with formal mathematical representation.

### A.2.2 THE CASE THAT FAIL TO COMPILE IN THE LEAN COMPILER

This part presents failure cases in automated formalization that fail Lean compilation, alongside comparative cases demonstrating successful compilation after CRAMF integration.

Figure 5: Baseline formalization failure case.

Figure 5 shows without retrieval augmentation of relevant concepts and definitions, the model attempted to formalize the concept of function continuity by inventing an identifier *Function.Continuous* based solely on its natural language interpretation. However, since this identifier does not exist in the Mathlib library, the compiler raised an error indicating that the symbol could not be resolved.

Figure 6: CRAMF-augmented success case.

Figure 6 illustrates that after incorporating the correct definition corresponding to function continuity, namely *ContinuousOn*, the model was able to generate a syntactically and semantically valid formalization based on the retrieved contextual information. As a result, the generated Lean code compiled successfully.

---

**Baseline Formalization**

NL-statement:

Prove that if $H$ is a normal subgroup of $G$ of prime index $p$, then for all $K \leqslant G$, either $K \leqslant H$ or $G = HK$ and $|K{:}K \cap H| = p$.

---

FL-statement:

theorem normal_of_index_is_prime {G : Type*} [Group G] {H K : Subgroup G}   (hH : H.Normal) (hG : Group.index H G = p)   : (K ≤ H ∨ G = H * K) ∧ (K ⊓ H).index K = p := sorry

---

Compilation error: unknown constant 'Group.index'

Figure 7: Baseline formalization failure case.

Figure 7 illustrates that due to the absence of necessary contextual knowledge or definitional support, the model failed to correctly reference the Mathlib implementation of the concept "index of a subgroup", and instead produced the undefined constant *Group.index*. Since this identifier does not exist in the library, the Lean compiler was unable to resolve it, resulting in a compilation failure.

---

**CRAMF-Augmented Formalization**

Context:

**"name"**: "Subgroup.index",
**"comments"**: " The index of a subgroup as a natural number. Returns `0` if the index is infinite. "
**"definition"**: "@[to_additive "The index of an additive subgroup as a natural number. Returns 0 if the index is infinite."] noncomputable def index : ℕ :=Nat.card (G ╱ H)"

---

FL-statement:

theorem exists_subgroup_of_index (G : Type*) [Group G] (n : ℕ) (hn : n | Nat.card G) :   ∃ **H : Subgroup G, H.index** = n := sorry

---

Compilation Result: Pass

Figure 8: CRAMF-augmented success case.

Figure 8 shows that with the support of the CRAMF framework, the model was provided with both the precise definition of the subgroup index, *Subgroup.index*, and accompanying semantic annotations. Consequently, during formalization, the model correctly referenced the well-defined constant *Subgroup.index*, enabling the compiler to resolve *H.index* and pass type checking.

### A.2.3 THE CASE THAT FAIL THE SEMANTIC CONSISTENCY TEST

This part presents cases where the formalization results pass Lean compilation but fail the semantic consistency test. We provide a detailed analysis of the underlying causes and further demonstrate the effectiveness of the CRAMF framework by comparing these results with those generated using CRAMF-enhanced inputs.

---

**Prompt for Rewrite mechanism**

NL-statement:

Prove that the improper integral $\int_a^{+\infty} \frac{dx}{x^p}$ converges when $p>1$, and diverges when $p\leq1$.

FL-statement:

theorem integral_one_div_rpow_extracted {a : ℝ} {p : ℝ} : 1 < p → ∫ (x : ℝ) in Set.Ioi a, 1 / x ^ p = 0 := sorry

Compilation result: pass

Back_translation:

Prove that $\int_a^{+\infty} \frac{dx}{x^p} = 0$ for 1 < p.

Semantic test result: different.

---

Figure 9: Baseline formalization failure case.

Figure 9 shows that without relevant contextual definitions, the model generated a Lean-compilable statement that failed to capture the full semantics of the original problem. It only addressed convergence for $p > 1$, omitted divergence for $p \leq 1$, and used the incorrect interval $(a, +\infty)$ instead of $(0, y)$. These omissions caused semantic inconsistency in the back-translation test.

---

**CRAMF-Augmented Formalization**

context:

**"name"**: "Filter.Tendsto",
**"comments"**: " 'Filter.Tendsto' is the generic 'limit of a function' predicate. 'Tendsto f $l_1$ $l_2$' asserts that for every '$l_2$' neighborhood 'a', the 'f' -preimage of 'a' is an '$l_1$' neighborhood."
**"definition"**: "def Tendsto (f : α → β) ($l_1$ : Filter α) ($l_2$ : Filter β) :=\n $l_1$.map f ≤ $l_2$"

FL-statement:

theorem integral_eq_intervalIntegral :
  (∀ p > 1, ∃ L : ℝ, **Tendsto** (λ y => ∫ x in (0)..y, 1/x^p) atTop ($\mathcal{N}$ L))
∧\n (∀ p ≤ 1, ¬ ∃ L : ℝ, **Tendsto** (λ y => ∫ x in (0)..y, 1/x^p) atTop ($\mathcal{N}$ L)) := by sorry

Compilation result: pass

Back-translation:

Prove that $\int_a^{+\infty} \frac{dx}{x^p}$ converges if $p > 1$ and diverges if $p \leq 1$ .

Semantic test result: same.

---

Figure 10: CRAMF-augmented success case.

Figure 10 shows the CRAMF-augmented formalization, which contrasts with the baseline result shown in Figure 9. By introducing the key concept of 'function limit' using the precise definition *Filter.Tendsto* and its semantic context, the model correctly expresses convergence behavior, distinguishes between the convergent case ($p > 1$) and divergent case ($p \leq 1$), and uses the appropriate interval integral $\int_{x \in (0,y)}$. This better captures the nature of the improper integral described in the original natural language. As a result, the formalization both compiles and preserves semantic alignment with the original problem.

---

**Baseline Formalization**

NL-statement:

Prove that the $\lim_{x \to x_0} c = c$ for any constant c.

---

FL-statement:

theorem const_lim {c : ℝ} {$x_0$ : ℝ} : (∀ ε > 0, ∃ δ > 0, ∀ x, |x - $x_0$| < δ → |c - c| < ε) := sorry

---

Compilation result: pass

---

Back_translation:

Prove that the constant sequence c converges to c.

---

Semantic test result: different.

Figure 11: Baseline formalization failure case.

---

**CRAMF-Augmented Formalization**

Context:

**"name"**: "Filter.Tendsto",
**"comments"**: " 'Filter.Tendsto' is the generic **'limit of a function'** predicate. 'Tendsto f $l_1$ $l_2$' asserts that for every '$l_2$' neighborhood 'a', the 'f' -preimage of 'a' is an '$l_1$' neighborhood."
**"definition"**: "def Tendsto (f : α → β) ($l_1$ : Filter α) ($l_2$ : Filter β) :=\n $l_1$.map f ≤ $l_2$"

---

FL-statement:

theorem tendsto_const_nhds_ext {c : α} {$x_0$ : α} [TopologicalSpace α] : **Filter.Tendsto (fun _x => c)** (nhds $x_0$) (nhds c) := sorry

---

Compilation Result: Pass

---

Back-translation:

Prove that the constant function $f(x) = c$ is continuous at any point $x_0$ .

---

Semantic test result: same.

Figure 12: CRAMF-augmented success case.

Figure 11 shows a different failure case. In the original natural language description, the limit is taken as $x \to x_0$, where the expression under consideration is a constant function $f(x) = c$. That is, the function is independent of $x$, and the goal is to prove that the limit of a constant function at any point equals the constant itself. However, in the model's translation, it misinterpreted the problem as concerning the limit of a sequence and failed to recognize the underlying mathematical concepts of "constant function" and "function limit." Consequently, the back-translation deviated semantically from the original statement, and the semantic test was not passed.

By contrast, Figure 12 illustrates that with the integration of the CRAMF framework, we identified the core mathematical concept of function limit" from the original proposition and explicitly provided its corresponding definition from the Mathlib library. With this contextual support, the model generated a correct formalization and was able to translate it back into the constant function is continuous at any point." Mathematically, continuity of a constant function at every point is equivalent to the existence of the function's limit at every point and that the limit equals the function value. Therefore, the semantic equivalence was preserved.

The above case studies demonstrate that the CRAMF framework not only enables accurate extraction of implicit mathematical concepts, but also effectively identifies their semantically aligned definitions in Lean 4. By providing essential contextual information to the autoformalization models, CRAMF significantly reduces both compilation failures and semantic inconsistencies.

## B  EXPERIMENT SETUP DETAILS

This section documents the precise hardware configurations and software dependencies used across experiments. We ensure reproducibility by specifying model parameters, runtime environments, and system specifications.

### B.1  MODEL PARAMETER CONFIGURATION

This section details the experimental inference configurations and software dependencies, as documented in Table 8 and Table 9.

Table 8: Inference configurations for all models.

| Model | Deployment Configuration | Sampling Parameters |
|---|---|---|
| Herald-7B | dtype="bfloat16" | temperature=0.7 max_tokens=1024 n=10 |
| Kimina-7B | tensor_parallel_size=1 | temperature=0.6 top_p=0.95 max_tokens=2048 n=10 repetition_penalty=1.2 |
| InternLM-Math-7B | dtype="bfloat16" tensor_parallel_size=1 | temperature=0.1 max_tokens=1024 |

### B.2  RUNTIME ENVIRONMENT CONFIGURATION

To ensure consistency and reproducibility of experimental results, all models were evaluated under the same hardware and software setup, as detailed below. All experiments were conducted on a server equipped with:

- **GPU:** 4×NVIDIA A800 80GB PCIe
- **GPU memory:** 80GB per card (total 320GB)
- **Operating mode:** PCIe Gen4, NVLink not enabled

- **CPU:** AMD EPYC 7R32 (128 cores / 256 threads)

- **System memory:** 1TB DDR4 ECC RAM

The software stack used for all experiments is summarized in the Table 9.

Table 9: Software versions and configuration used in the experiments.

| Category | Version / Configuration |
|---|---|
| Operating System (Ubuntu) | 22.04 LTS (Kernel 6.5.0-44-generic) |
| GPU Driver (NVIDIA) | 535.171.04 |
| CUDA | 12.2 |
| Python | 3.10.12 |
| PyTorch | 2.1.0 |
| vLLM | 0.3.2 |
| Transformers | 4.36.2 |
| Lean Compiler | 4.20.0-rc5 (x86_64-linux-gnu) |
| Elan Version Manager | 4.1.2 |
| Mathlib4 | leanprover/lean4:v4.20.0-rc5 |

## C PROMPT

This section details all prompt templates employed for core tasks, including problem rewriting, concept extraction, concept analysis, autoformalization, back-translation, and semantic scoring. Each prompt is explicitly structured to guide LLMs in formalizing mathematical problems or analyzing concepts under strict output constraints.

### C.1 PROMPT FOR REWRITE MECHANISM

For mathematical problems lacking explicitly stated concepts, we employ the DeepSeek-R1 model to perform mathematical modeling and problem rewriting via the following structured prompt template, rendering them amenable to formalization:

> **Prompt for Rewrite mechanism**
>
> You are a mathematics expert specializing in mathematical modeling and analysis of applied mathematical problems (e.g., combinatorial mathematics).
>
> When given a user's problem description:
> 1. Perform mathematical modeling by rewriting the problem to explicitly express all involved mathematical concepts . (e.g., sets, functions, equations, probability)
> 2. Preserve the core requirements of the original problem.
> 3. Use formal mathematical language in English.
> 4. Output ONLY the rewritten problem text without any explanations or additional content.
>
> Rewrite the following mathematical problem, explicitly formalizing all mathematical concepts:
> {problem}

Figure 13: Prompt for Rewrite mechanism.

### C.2 PROMPT FOR BACK-TRANSLATION

The prompt for reverse-engineering Lean4 formal specifications into natural language descriptions using the InternLM-Math-7B model is provided below.

> **Prompt for Back-translate**
>
> [UNUSED_TOKEN_146]user
> Convert the formal statement into natural language:
>
> ```lean
> {code}
> ```

Figure 14: Prompt for Rewrite mechanism

### C.3 PROMPT FOR CONCEPT ANALYSIS

We use DeepSeek-V3 to perform concept analysis on existing mathematical concepts, with the following prompt template:

> **Prompt for Concept analysis**
>
> You are a mathematics expert specialized in mathematical terminology and concise explanations. Provide a professional one-sentence explanation for the following mathematical concept. Requirements:
>
> 1. Explanation must be concise while preserving key information.
> 2. Use standard mathematical terminology.
> 3. Output ONLY the English explanation without any additional content.
>
>  Example format:
> Input: maximal value
> Output: A local maximum of a function at a point a is its value f(a) that is ≥ all other f(x) within some neighborhood of a.
>
> Concept to explain:
> {concept}

Figure 15: Prompt for Concept analysis.

### C.4 PROMPT FOR CONCEPT EXTRACTION

For concept extraction from mathematical definitions or problems when building knowledge bases, we use DeepSeek-V3 with this prompt:

> ### Prompt for Concept extraction
>
> Analyze the given mathematical theorem description and:
> 1. Extract fundamental mathematical concepts (e.g., "derivative", "integral", "vector space", "group theory").
> 2. Identify the most relevant mathematical domain (e.g., "Calculus", "Algebra", "Geometry").
> Return JSON format: {"concepts": ["...", ...], "domain": "..."}
>
> Analyze this mathematical theorem description:
> {description}

Figure 16: Prompt for Concept extraction.

## C.5    PROMPT FOR SEMANTIC SCORING

To evaluate retriever performance, we score semantic relevance between definitions and mathematical problems using deepseek-R1.

> ### Prompt for Semantic relevance scoring
>
> You are a mathematical relevance assessment expert. Your task is to evaluate the semantic relationship between mathematical definitions and problem statements based on strict scoring criteria. Provide JSON-formatted output with score and rationale.
>
>  For the problem statement:
> {problem_statement}
>
> Evaluate the relevance of this definition:
> {mathematical_definition}
>
> Scoring criteria:
> 1. 2 points (strong relevance):
>    Demonstrates direct semantic connection to the problem domain.
>    Exhibits clear mathematical significance.
>
> 2. 1 point (weak relevance):
>    Shows thematic relationship to problem domain.
>    Lacks direct semantic linkage.
>    Lacks formal mathematical utility.

Figure 17: Prompt for Semantic scoring.

## C.6    PROMPT FOR AUTOFORMALIZATION

The prompt for automated formalization in the Hearld and Kimina models is shown below.

```
Prompt for Autoformalization

[UNUSED_TOKEN_146]user
Task: Convert following problem into LEAN 4:

{natural_language_statement}
[UNUSED_TOKEN_145]

[UNUSED_TOKEN_146]assistant
Here is the formal statement in LEAN 4:
```lean
theorem
```

Figure 18: Prompt for Autoformalization.

### C.7 PROMPT FOR AUTOFORMALIZATION WITH CRAMF

The prompt augmented with pertinent definitional information is presented below.

```
Prompt for Autoformalization with CRAMF

[UNUSED_TOKEN_146]user
Task: Convert the following problem into LEAN 4 code. Use
the mathematical definitions provided below for reference,
but DO NOT translate the definitions themselves.

Problem to translate:
{problem}

Reference definitions (for context only):
{context_entries}

[UNUSED_TOKEN_145]

[UNUSED_TOKEN_146]assistant
Here is the formal statement in LEAN 4:
```lean4
theorem
```

Figure 19: Prompt for Autoformalization with CRAMF.

## D LARGE LANGUAGE MODEL USAGE

This paper was prepared with the assistance of a large language model. Our use of the LLM was strictly limited to the linguistic polishing of the manuscript. The process was as follows:

- We completed a full draft of the paper, containing all key contributions.
- Selected passages where the phrasing was perceived to be unclear or inelegant were identified.
- These passages were input into the LLM with instructions to suggest alternative phrasings for improved clarity and flow, while strictly preserving the original technical meaning.
- We critically reviewed, modified as necessary, and approved every suggestion before its incorporation into the final manuscript.

The LLM played no role in the conception of the research, the formulation of hypotheses, the design and execution of experiments, data analysis, or the interpretation of results. Its function was analogous to that of a sophisticated proofreading tool. We hereby affirm that they are solely responsible for the entire scientific content and the accuracy of all statements in this work.

# E    ANALYSIS OF SYSTEM LATENCY AND CONTEXT BUDGET

We systematically examined the performance of the CRAMF framework in terms of system latency and context budget, aiming to comprehensively evaluate its efficiency and scalability in practical deployment.

## E.1    DELAY ANALYSIS OF EACH MODULE OF THE SYSTEM

We conducted a fine-grained breakdown of the end-to-end latency of CRAMF under a batch size of 1, using a hardware setup of $4 \times$ NVIDIA A800 80GB GPUs. The results are summarized in the Table 10. We report the mean, standard deviation, minimum, and maximum latency of each CRAMF module to comprehensively characterize their delay distribution and performance variability.

Table 10: Latency Breakdown of Individual Modules in the CRAMF Framework.

| Module | Mean(ms) | Std(ms) | Min(ms) | Max(ms) |
|---|---|---|---|---|
| Concept Extraction & Parsing | 2261.63 | 181.26 | 2002.48 | 2903.49 |
| Symbol-Level Keyword Matching | 124.18 | 35.46 | 103.20 | 191.17 |
| Semantic Similarity Retrieval | 363.16 | 20.14 | 326.89 | 407.95 |
| Rerank | 14.40 | 47.43 | 7.69 | 485.81 |
| Generation–Herald-7B | 1752.11 | 295.40 | 1070.58 | 2220.34 |
| Generation–GPT-4o | 9621.92 | 1080.51 | 8414.50 | 11440.51 |
| Generation–Kimina-7B | 7633.74 | 5234.67 | 4118.43 | 19675.50 |
| Generation–DeepSeek-V3 | 6903.24 | 444.06 | 6443.82 | 7998.04 |

Table 11: End-to-End Latency of CRAMF Integrated with Different Generation Models.

| Model | End-to-End Latency (ms) |
|---|---|
| Herald-7B | 4515.48 |
| GPT-4o | 12385.29 |
| Kimina-7B | 10397.11 |
| DeepSeek-V3 | 9666.61 |

## E.2    CONTEXT BUDGET ANALYSIS

We further count the contextual token usage of each model on different datasets before and after using CRAMF enhancement, as shown in Table 12. The results show that although CRAMF does increase the context length, the maximum token usage is within 4.3 k, which is completely within the context window bearing range of the current LLMs.

Table 12: Token usage for each model and dataset before and after using CRAMF enhancement.

| Dataset | Condition | Min Tokens | Max Tokens |
|---|---|---|---|
| AdvancedMath | Base | 140 | 386 |
| | +CRAMF | 253 | 4,256 |
| MiniF2F | Base | 136 | 299 |
| | +CRAMF | 222 | 1,310 |
| ProofNet | Base | 131 | 281 |
| | +CRAMF | 401 | 4,085 |

Through reasonable modular design, CRAMF achieves a significant improvement in formal accuracy and achieves a good balance of efficiency and effectiveness under the premise of introducing limited time delay and controllable context overhead.

