# OpenReview forum: "Automated Formalization via Conceptual Retrieval-Augmented LLMs"
_ICLR.cc/2026/Conference — ICLR 2026 Poster_

### Official Review · Reviewer_AM7R · 2025-10-26

**Soundness:** 3
**Presentation:** 3
**Contribution:** 3
**Rating:** 6
**Confidence:** 3

**Summary:**

This paper focuses on the task of automatic formalization in theorem proving, which currently faces two major challenges: model hallucination and the semantic gap caused by ambiguous or missing premises in natural language descriptions. To address these issues, the authors propose a framework called CRAMF (Concept-driven Retrieval-Augmented Mathematical Formalization). The framework consists of three main components: (1) construction of a concept definition knowledge base, (2) mathematical concept extraction, and (3) definition retrieval. Experimental results demonstrate that CRAMF can effectively extract information useful for formalization, thereby improving overall accuracy.

**Strengths:**

The paper is very well-written, easy to follow and well-organized.

This paper introduces a framework for automatically constructing a concept-definition knowledge base based on Mathlib4, addressing the problem of a lack of structured knowledge bases in the automated formalization domain.

This paper introduces the CRAMF-Guided Autoformalization process, which first extracts math concepts from question and then retrieve definitions from pre-defined knowledges. Then, based on the informal statement and definition, realize autoformalization, thus improving the accuracy.

Experiments on different datasets show the effectiveness of the CRAMF framework. Through the ablation study, it also shows the reasonability of each module in this framework.

**Weaknesses:**

Although the CRAMF method substantially improves both retrieval accuracy and auto-formalization performance, I still have some concerns.

The paper uses MathBERT for semantic encoding, but it is not clear whether the authors have compared it with other encoding models that might achieve better performance.

In addition, regarding the baselines, there already exist several retrieval approaches for Lean based on natural language, such as LeanSearch and LeanExplore. It would strengthen the paper if comparisons with these methods were included.

**Questions:**

Please refer to the Weakness section.

---

> ### Author Response · Authors · 2025-11-28
> **Response to Reviewer  AM7R**
>
> We thank the reviewers for their careful reading and constructive feedback. We appreciate the positive assessments regarding the clarity of our presentation, the novelty of the concept-definition knowledge base, and the effectiveness of the CRAMF framework. The concerns raised about semantic encoding choices and comparisons with Lean-specific retrievers are valuable, and we have conducted additional analyses and experiments to address them. Our detailed responses are provided below.
>
> **Weakness1: Lack of basis for MathBERT encoder selection.**
>
> We thank the reviewers for their affirmation and suggestions. We chose MathBERT as the semantic encoder based on its unique advantages brought by pre-training in the mathematical field. The model is trained on a large number of mathematical texts (such as papers, textbooks, and online resources) and formulas, allowing it to accurately capture the semantic associations and polymorphisms of mathematical concepts (such as understanding the different meanings of "neighborhood" in topology and metric spaces). Compared with general-purpose domain encoders, MathBERT is more suitable for bridging the semantic gap between natural language mathematical descriptions and formal definitions, so it is the theoretically optimal choice for our task.
>
> **Weakness2: CRAMF lacks lean-related retrieval approaches contrasts.**
>
> We sincerely thank the reviewer for raising this important point regarding comparisons with specialized formal retrievers. In response, we have now included a comprehensive comparison against two Lean-specific retrieval engines—LeanSearch \[1\] and LeanExplore \[2\]—which are explicitly designed for semantic search over Lean 4 declarations and Mathlib4. As shown in the updated Tables 1 and 2, CRAMF consistently outperforms both systems in Average Contribution Score (ACS) and HitRate@3, demonstrating its stronger capability in retrieving accurate and relevant definitions in real formalization scenarios. Relevant results have been added to the paper.
>
> Table 1: Average contribution scores (ACS) of retrieval methods on three datasets.
>
> |     |     |     |     |
> | --- | --- | --- | --- |
> | Method | MiniF2F | ProofNet | AdvancedMath |
> | LeanSearch | 1.58 | 1.63 | 1.56 |
> | LeanExplore | 1.42 | 1.51 | 1.35 |
> | CRAMF | 2.07 | 2.14 | 1.93 |
>
>
> Table 2: Top-3 hit rate of relevant definitions (HitRate@3) for each retrieval framework across three datasets.
>
> |     |     |     |     |
> | --- | --- | --- | --- |
> | Method | MiniF2F | ProofNet | AdvancedMath |
> | LeanSearch | 36.8% | 42.3% | 35.1% |
> | LeanExplore | 31.5% | 37.7% | 23.8% |
> | CRAMF | 44.2% | 50.6% | 42.9% |
>
> [1]  Gao, Guoxiong, et al. "A semantic search engine for Mathlib4." arXiv preprint arXiv:2403.13310 (2024).
>
> [2]  Asher, Justin. "LeanExplore: A search engine for Lean 4 declarations." arXiv preprint arXiv:2506.11085 (2025).

---

### Official Review · Reviewer_bPuu · 2025-10-29

**Soundness:** 3
**Presentation:** 2
**Contribution:** 2
**Rating:** 4
**Confidence:** 4

**Summary:**

This paper proposes CRAMF, a concept-driven retrieval-augmented framework for automated statement formalization in Lean 4. It builds a concept–definition knowledge base from Mathlib4 and uses hybrid retrieval with reranking to provide context for LLMs. Experiments on miniF2F, ProofNet, and a new AdvancedMath dataset show improved compilation and formalization accuracy across models.

**Strengths:**

- The paper applies RAG to statement formalization tasks, and the retrieved contextual information effectively improves model performance.
- Code and benchmark data are provided in supplementary material, supporting reproducibility.

**Weaknesses:**

- Lack of novelty. It is already well established that RAG improves performance on knowledge-intensive tasks, and similar ideas have been explored in auto theorem proving and auto formalization in Lean, e.g., ReProver [1], LEGO Prover [2], and recently Hilbert [3]. It is unsurprising that RAG also works in statement formalization. Moreover, key components such as reverse translation of Mathlib4 entities, query enhancement, and dual-pathway hybrid retrieval have already been used in Mathlib retrieval systems like LeanSearch [4] and LeanExplorer [5].
- The range of models evaluated is limited. More recent and stronger models such as Gemini-2.5-pro, GPT-5, Claude-4.1, and Goedel-Formalizer-V2 (8B/32B) should be included for comparison. Their inclusion would strengthen the experimental section. It should also be noted that Kimina-7B is trained on miniF2F and ProofNet, making comparisons on these benchmarks less fair.
- Formalization accuracy is evaluated only through automated checks, without human expert verification, which weakens the reliability of the results.
- More explanation is needed for the AdvancedMath dataset. Although it is included in the supplementary material, its motivation and basic characteristics should be summarized in the main text. It seems all problems come from a single source and focus on calculus, which should be clarified.

[1] Yang, Kaiyu, et al. "Leandojo: Theorem proving with retrieval-augmented language models." Advances in Neural Information Processing Systems 36 (2023): 21573-21612.

[2] Wang, Haiming, et al. "Lego-prover: Neural theorem proving with growing libraries." arXiv preprint arXiv:2310.00656 (2023).

[3] Varambally, Sumanth, et al. "Hilbert: Recursively Building Formal Proofs with Informal Reasoning." arXiv preprint arXiv:2509.22819 (2025).

[4] Gao, Guoxiong, et al. "A semantic search engine for Mathlib4." arXiv preprint arXiv:2403.13310 (2024).

[5] Asher, Justin. "LeanExplore: A search engine for Lean 4 declarations." arXiv preprint arXiv:2506.11085 (2025).

**Questions:**

- Does “Formalization Accuracy Rate@10” in Table 2 refer to the probability that at least one of the ten outputs passes the semantic consistency check, or to the proportion of successful outputs among the ten? It would also be helpful to clarify whether the metric is calculated on all runs or only those that pass compilation.
- The notion of “concept” is somewhat vague. What exactly is its relationship with Mathlib definitions, classes, or structures? Is it a one-to-one mapping, or a more abstract layer? How do “concepts” influence the retrieval process?

---

> ### Author Response · Authors · 2025-11-25
> **Response to Reviewer bPuu**
>
> We sincerely appreciate your insightful feedback and the time you have dedicated to reviewing our paper. We have carefully addressed all the points raised in your review and provided detailed responses below. Should any aspects require further clarification, we would be glad to discuss them additionally. We hope our revisions and responses have adequately resolved your concerns, and we would be grateful if you could reconsider your overall rating of our work.
>
> **Regarding the concern raised in Weakness 1, It can be split into two aspects, which we will address point by point:**
>
> ## Weakness 1-1
> At present, there are methods used by RAG in automatic theorem proof. Compared with this, what are the innovative and unique challenges of using RAG in automatic formalization?
>
> ## Response
>
> There are essential differences in task objectives and semantic granularity between automatic formalization (AF) and automatic theorem proof (ATP), which brings new challenges and innovation space for the application of RAG. The core challenge of AF lies in identifying key mathematical concepts from semantically ambiguous and expressively diverse natural languages and accurately mapping them into unique normalized definitions; However, the input of ATP itself is formal content, and the retrieval target is the related theorem or proof step under the known premise. Therefore, the RAG system in AF needs to specially design an indexing and retrieval mechanism oriented to natural language mathematical concepts to solve the semantic gap and structured mapping, which are the core problems that do not exist in ATP. Our method aims at this challenge to propose a new retrieval strategy, so as to achieve effective knowledge enhancement in AF tasks.
>
> We use ReProver [1] as an example to illustrate their differences. Consider the mathematical problem:
>
> > "Prove that if a sequence xn converges, then the sequence xn must be bounded."
>
> For this problem, the AF task first needs to focus on the key semantics in the natural language mathematical problem, namely the three mathematical concepts `sequence`, `convergence`, and `boundedness`, while also effectively identifying implicit mathematical concepts and entities, such as the mathematical concept `limit` implied by the term `convergence`. It then retrieves the corresponding definitions in Mathlib for these core mathematical concepts, such as `Filter.Tendsto`, `Filter.atTop`, etc.
>
> The formalization and proof of this theorem in Lean 4 are as follows:
>
> ```lean
> theorem convergent_seq_bounded {x : ℕ → ℝ} (h : ∃ L, Filter.Tendsto x Filter.atTop (nhds L)) :
>         ∃ M, ∀ n, |x n| ≤ M := by
>   rcases h with ⟨L, h⟩
>   have h1 : ∀ ε > 0, ∃ N : ℕ, ∀ n ≥ N, |x n - L| < ε := by
>     intro ε hε
>     have := Metric.tendsto_atTop.mp h ε hε
>     exact this
>   ...
> ```
> In contrast, ReProver operates based on a given formal proof state, for example:
> ```lean
> x : ℕ → ℝ
> L : ℝ
> h : Tendsto x atTop (N L)
> ε : ℝ
> hε : ε > 0
> ⊢ ∃ N, ∀ n ≥ N, |x n - L| < ε
> ```
> It retrieves theorems that are helpful for advancing the proof goal, such as  `theorem Metric.tendsto_atTop `, to apply the proof tactic have, achieving the next proof state:
> ```lean
> x : ℕ → ℝ
> L : ℝ
> h : Tendsto x atTop (N L)
> ε : ℝ
> hε : ε > 0
> this : ∃ N, ∀ n ≥ N, Dist.dist (x n) L < ε
> ⊢ ∃ N, ∀ n ≥ N, |x n - L| < ε
> ```
> Finally, to address the unique challenges of autoformalization, this work specifically targets the semantic gap caused by conceptual polymorphism. We constructed a concept-definition knowledge base tailored for formal translation and introduced a novel, concept-driven retrieval mechanism. Through context-aware query augmentation and dual-channel hybrid retrieval, our approach accurately captures mathematical concepts across varying abstraction levels, thereby effectively bridging the semantic gap between natural language descriptions and formal definitions.
>
> [1] Yang, Kaiyu, et al. "Leandojo: Theorem proving with retrieval-augmented language models." Advances in Neural Information Processing Systems 36 (2023): 21573-21612.

---

> ### Author Response · Authors · 2025-11-25
> **Response to Reviewer bPuu**
>
> **Weakness 1-2: A comparison between the core components of the CRAMF framework and existing Mathlib retrieval systems.**
>
> Response:
>
> We admit that existing works (such as LeanSearch\[2\], LeanExplore\[3\]) have introduced techniques such as query enhancement in Mathlib retrieval, but CRAMF has essential differences in problem location and technical route. Take query enhancement as an example: LeanSearch converts queries into detailed descriptions containing informal and formal statements, aiming to improve query clarity to improve embedding matching, while CRAMF focuses on solving the semantic polymorphism of mathematical concepts and injects domain knowledge through concept parsing to bridge the semantic gap between natural language and formal definitions.
>
> In order to objectively verify the effectiveness of CRAMF in formal tasks, we have included the two in the retrieval performance comparison. As shown in Tables 1 and 2, CRAMF is significantly better than the baseline in ACS and HitRate@3, indicating that it has stronger definition retrieval capabilities in real formal scenarios. Relevant results have been added to the paper.
>
> Table 1: Average contribution scores (ACS) of retrieval methods on three datasets.
>
> | Method       | MiniF2F | ProofNet | AdvancedMath |
> |--------------|---------|----------|--------------|
> | LeanSearch   | 1.58    | 1.63     | 1.56         |
> | LeanExplore  | 1.42    | 1.51     | 1.35         |
> | CRAMF        | **2.07**| **2.14** | **1.93**     |
>
> Table 2: Top-3 hit rate of relevant definitions (HitRate@3) for each retrieval framework across three datasets
>
> | Method       | MiniF2F | ProofNet | AdvancedMath |
> |--------------|---------|----------|--------------|
> | LeanSearch   | 36.8%   | 42.3%    | 35.1%        |
> | LeanExplore  | 31.5%   | 37.7%    | 23.8%        |
> | CRAMF        | **44.2%**| **50.6%**| **42.9%**    |
>
>
> **For Weakness2：it can be divided into the following two points to respond.**
>
> **Weakness2-1: The evaluation model is limited in scope and fails to include some newer, stronger models.**
>
> In the experimental stage, our experiments have covered the most representative SOTA models at that time. We strongly agree with your suggestion to expand the scope of the model, and have conducted important supplementary experiments accordingly. We newly added a recent general model with strong performance and a model dedicated to automatic formalization as baselines for comparative experiments, and selected three models: Gemini2.5-flash, Claude4.1 and Godel-Formalizer-V2-8B. The Compilation Pass Rate @ 10 and Formalization Accuracy Rate @ 10 indicator evaluation results of these models before and after joining the CRAMF framework are shown in Tables 3 and 4.
>
> Table 3: A comparison of the Compilation Pass Rate@10 for the three additional baseline models versus their CRAMF-enhanced counterparts across three datasets. RG (Relative Gain) indicates the relative improvement rate.
>
> |     |     |     |     |     |     |     |     |     |     |
> | :--- | :---: | :---: | :---: | :---: | :---: | :---: | :---: | :---: | :---: |
> |**Model** |       |   **MiniF2F** |     |      |  **Proofnet**   |     |        | **AdvancedMath**    |     |
> |     | Base | +CRAMF | RG  | Base | +CRAMF | RG  | Base | +CRAMF | RG  |
> | Gemini2.5-flash | 82.38% | 89.34% | +8.45% | 70.86% | 75.13% | +6.03% | 79.27% | 82.08% | +3.54% |
> | Claude4.1 | 95.08% | 97.96% | +3.03% | 81.28% | 85.56% | +5.27% | 98.84% | 99.42% | +0.59% |
> | Godel-Formalizer-V2-8B | 99.59% | 100% | +0.41% | 85.56% | 87.43% | +2.14% | 98.84% | 100.00% | +1.17% |
>
> Table 4: A comparison of the Formalization Accuracy Rate@10 for the three additional baseline models versus their CRAMF-enhanced counterparts across three datasets. RG (Relative Gain) indicates the relative improvement rate.
>
> |     |     |     |     |     |     |     |     |     |     |
> | :--- | :---: | :---: | :---: | :---: | :---: | :---: | :---: | :---: | :---: |
> |**Model** |       |   **MiniF2F** |     |      |  **Proofnet**   |     |        | **AdvancedMath**    |     |
> |     | Base | +CRAMF | RG  | Base | +CRAMF | RG  | Base | +CRAMF | RG  |
> | Gemini2.5-flash | 72.13% | 81.97% | +13.64% | 56.95% | 63.10% | +10.80% | 43.93% | 50.87% | +15.80% |
> | Claude4.1 | 83.60% | 92.62% | +10.79% | 72.19% | 79.68% | +10.38% | 57.23% | 65.90% | +15.15% |
> | Godel-Formalizer-V2-8B | 88.11% | 95.08% | +7.91% | 72.99% | 80.21% | +9.90% | 57.80% | 68.21% | +18.01% |
>
>
> [2]  Gao, Guoxiong, et al. "A semantic search engine for Mathlib4." arXiv preprint arXiv:2403.13310 (2024).
>
> [3]  Asher, Justin. "LeanExplore: A search engine for Lean 4 declarations." arXiv preprint arXiv:2506.11085 (2025).

---

> ### Author Response · Authors · 2025-11-25
> **Response to Reviewer bPuu**
>
> **Weakness2-2: The Kimina-7B model is trained on two benchmark datasets, miniF2F and ProofNet, so it is unfair to compare with other models on this test set.**
>
> The core goal of this work is to verify the generic enhancement capabilities of CRAMF, rather than directly performing horizontal comparisons between models. We note that although Kimina-7B may have been exposed to partial data during training, its baseline performance (65.0% accuracy on ProofNet) is far from saturated. This suggests that the model does not fully "memorize" the test samples. In this case, CRAMF can still significantly improve its performance (to 69.3%), proving that the framework can help the model transcend the incomplete memory of its initial training by injecting structured knowledge, and achieve more robust and accurate code generation, reflecting the unique value of the method itself.
>
> **Weakness3：The formal assessment relies on automated evaluation, lacking human expert judgment.**
>
> Firstly, the automated evaluation process employed in this paper is the current mainstream method for evaluating the automatic formalization, such as in Leanworkbook \[4\], Hearld \[5\], and Kimina \[6\]. Nonetheless, we acknowledge that incorporating verification by human experts would further enhance the credibility of the experimental results. Therefore, we supplemented the automated evaluation by incorporating human expert verification and analyzed the correlation and agreement between automated and human evaluations using metrics including **Cohen's Kappa coefficient**. The sample sizes across datasets and the consistency evaluation metrics are presented in Table 5. The results show a Cohen's Kappa of **0.903** and an accuracy of **0.955**, indicating substantial agreement between automated and human evaluations and demonstrating the reliability of the automated evaluation method.
>
> Table 5: Experimental results showing sample sizes and agreement evaluation metrics for the MiniF2F, ProofNet, and AdvancedMath datasets.
>
> |     |     |     |     |     |     |     |
> | :--- | :---: | :---: | :---: | :---: | :---: | :---: |
> | Dataset | Sample Size | Cohen's Kappa | Accuracy | Precision | Recall | F1 Score |
> | MiniF2F | 48  | 0.957 | 0.979 | 0.952 | 1.000 | 0.976 |
> | Proofnet | 74  | 0.887 | 0.946 | 0.931 | 0.931 | 0.931 |
> | AdvancedMath | 34  | 0.946 | 0.941 | 0.857 | 0.857 | 0.857 |
> | overall | 156 | 0.903 | 0.955 | 0.930 | 0.946 | 0.938 |
>
> The following is the experimental procedure:
>
> **Sample Selection**
>
> We randomly selected 10% of the generated results from each dataset, covering all baseline models and CRAMF versions. For each (problem, model) pair, we selected the "best" output from the automated evaluation – specifically, the first output that passed both compilation and semantic consistency checks – for human assessment. If no output passed both checks, we selected the first output that compiled (if any), otherwise, the first generated output was chosen. This sampled data constituted the evaluation set for this experiment.
>
> **Evaluation Procedure**
>
> We asked three mathematics experts with experience in using Lean/Mathlib to conduct a double-blind assessment, and the assessors were unaware of the sample source and the automated assessment results. The assessment used a binary labeling scheme: "Correct" or "Incorrect". "Correct" required the formal code to fully and accurately capture the mathematical meaning of the original natural language statement, including all key assumptions, conclusions, and semantic nuances. "Incorrect" indicated the formal code contained errors, omissions, or misleading elements.
>
> **Automated Evaluation Labels**
>
> The automated labels were based on the "Formalization Accuracy Rate@10" (FAR@10) defined in the paper. For each problem, the automated label was "Correct" if at least one output passed both the compilation and semantic consistency checks; otherwise, it was "Incorrect". For the sampled problems, we recorded the automated labels for both the baseline and CRAMF model scenarios to enable comparison with the human evaluation results.
>
> **Agreement Analysis Metrics**
>
> We selected Cohen's Kappa coefficient to measure the agreement between the automated and human evaluations, accounting for the probability of random agreement. It is calculated as:
>
> Kappa = (Po - Pe) / (1 - Pe)
>
> where Po is the observed agreement proportion and Pe is the expected agreement proportion. The interpretation standards for the Kappa coefficient are: <0.20 (Slight agreement), 0.21-0.60 (Moderate agreement), 0.61-0.80 (Substantial agreement), and 0.81-1.00 (Almost perfect agreement). Additionally, we also used supplementary metrics including Accuracy, Precision, Recall, and F1-score to comprehensively evaluate the reliability of the automated evaluation system.

---

> ### Author Response · Authors · 2025-11-25
> **Response to Reviewer bPuu**
>
> **Weakness4：More explanation is needed for the AdvancedMath dataset.**
>
> Reviewers are appreciated for their comments. We have supplemented the paper with a detailed description of the AdvancedMath dataset. The AdvancedMath data set is selected from the authoritative textbook "Advanced Mathematics (8th Edition)" published by Tongji University, which ensures the standardization and mathematical rigor of the title expression. The data set contains a total of 173 questions. Besides covering the core content of calculus, **it also systematically includes important branches such as spatial analytic geometry, vector algebra, series theory and ordinary differential equations.**
>
> In terms of difficulty and complexity, compared with miniF2F, which focuses on elementary mathematics and competition questions and focuses on logical reasoning, and ProofNet, which has higher difficulty and knowledge density, **AdvancedMath involves moderate complexity of problems, and integrates logical reasoning, symbolic operation and natural language understanding.** It is suitable as a benchmark for evaluating the comprehensive performance of formal translation systems.
>
> **Question1：The definition of the Formalization Accuracy Rate@10**
>
> Regarding the explanation of Formalization Accuracy Rate@10: For each mathematical problem, we generate 10 Lean4 translations in parallel. If at least one of these translations passes both the compilation and semantic tests, we consider the translation attempt for that problem to be accurate and successful. The Formalization Accuracy Rate@10 represents the proportion of problems in a dataset for which this successful translation occurs.
>
> **Question2：The ontological definition of "concepts," their precise relationship with Mathlib definitions, and their operational role in the retrieval process.**
>
> The relationship between concepts and Mathlib definitions (including def, structure, class, etc.) is one-to-many, meaning a single mathematical concept can correspond to multiple Mathlib definitions. This point can be found in Section 2.1.1 of the paper.
>
> How Mathematical Concepts Influence the Retrieval Process?
>
> First, mathematical concepts are used to enhance user queries. When we input mathematical problems to the retriever, we don't directly vector embedding the problems, because the semantic information contained in them is complicated. The definitions we want to retrieve are most related to the mathematical concepts, so we use large models to extract the mathematical concepts in the problem to accurately condense the key semantics of the original mathematical problem, reduce the retrieval granularity and make the semantics of the query more accurate. At the same time, we generate the technical term explanation of the concept, and add rich mathematical field and application background information to the concept, so as to match the most relevant definition information at the semantic granularity and abstract level in the retrieval stage (abstract level can be understood as different abstract spaces in mathematics, such as topological space, metric space, etc. Semantic granularity can be understood as the level of detail or accuracy used in describing mathematical objects, such as function continuity, function continuity at a certain point, function continuity at a certain interval).
>
> Secondly, mathematical concepts drive the dual-channel retrieval process. On the one hand, we extract concisely defined search keys from mathematical concepts, and regularly match related definition symbol names. On the other hand, using "concept + term explanation" as a query, the semantic similarity between it and the existing concepts in the knowledge base is calculated, and the definition information of similar concepts is retrieved.
>
> Finally, the concept supports reordering and disambiguation. After a plurality of candidate definitions are preliminarily retrieved, the system uses a reordering model to evaluate the semantic matching degree of each definition with the query concept. This process relies on semantic alignment between natural language interpretations of concepts and candidate definition annotations, further screening out the most relevant definitions.
>
> [4]  Ying, Huaiyuan, et al. "Lean workbook: A large-scale lean problem set formalized from natural language math problems." Advances in Neural Information Processing Systems 37 (2024): 105848-105863.
>
> [5]  Gao, Guoxiong, et al. "Herald: A natural language annotated lean 4 dataset." arxiv preprint arxiv:2410.10878 (2024).
>
> [6]  Wang, Haiming, et al. "Kimina-prover preview: Towards large formal reasoning models with reinforcement learning." arxiv preprint arxiv:2504.11354 (2025).

---

### Official Review · Reviewer_UEGf · 2025-10-31

**Soundness:** 2
**Presentation:** 2
**Contribution:** 3
**Rating:** 6
**Confidence:** 3

**Summary:**

This paper presents CRAMF, a retrieval-augmented framework that enhances LLM-based automated formalization for theorem proving in Lean 4. CRAMF builds a structured concept-definition knowledge base from Mathlib4 (over 26,000 definitions and 1,000 core concepts) and retrieves relevant definitions to reduce hallucinations and semantic gaps during autoformalization. Experiments on miniF2F, ProofNet, and a new dataset show consistent improvements, with up to 62.1% accuracy gain.

**Strengths:**

1. The paper precisely identifies two major obstacles in automated formalization—model hallucination and the semantic gap arising from ambiguous or incomplete natural language premises.
2. This paper introduces a concept-level retrieval-augmented generation (RAG) mechanism and builds a concept–definition knowledge base from Mathlib4.
3. This paper demonstrates through experiments on miniF2F, ProofNet, and the newly proposed AdvancedMath benchmark that CRAMF can be seamlessly integrated into LLM-based autoformalization pipelines, consistently enhancing translation accuracy

**Weaknesses:**

1. The paper is primarily system-oriented and lacks formal justification or complexity analysis of why the hybrid retrieval pipeline improves semantic grounding. For example, the mapping from informal expressions to formal concepts could benefit from a more rigorous evaluation of ontology precision and recall.
2. The experimental comparison focuses mainly on RAG-style retrievers (e.g., BM25, HyDE), without including other autoformalization-specific retrieval systems
3. While the retrieval strategy is well adapted to formal reasoning, its methodological foundation largely follows existing RAG frameworks. The paper’s novelty lies primarily in domain adaptation to mathematical formalization rather than introducing fundamentally new retrieval theory or mechanisms.

**Questions:**

Please refer to the Weakness section.

---

> ### Author Response · Authors · 2025-11-28
> **Response to Reviewer UEGf**
>
> Thanks to the reviewers for their meticulous review and valuable comments on our work. We have carefully analyzed and supplemented your questions. The following is our response to each comment one by one:
>
> **Weakness1: The innovation of this paper lies more in its adaptation in the field of mathematical formalization rather than in proposing a fundamentally new retrieval theory or mechanism.**
>
> Thanks to the reviewers for their valuable comments. We fully agree that the overall RAG framework has a mature foundation in the general field. However, the general RAG framework is not very suitable for the highly specialized, semantically sensitive and symbol-intensive task of automatic formalization. The fundamental reason is that the existing autoformalization task faces three problems that the general RAG cannot solve, and our method can solve it well:
>
> Mathematical concept polymorphism problem. The same natural language concept corresponds to completely different formal definitions in different mathematical contexts. Universal RAG relies on semantic similarity retrieval, which cannot distinguish the formal definitions of the same concept at different levels of abstraction (such as topological space and metric space), resulting in semantic inaccurate retrieval results and typing errors. We construct a structured concept-definition knowledge base and design context-aware query enhancements. By letting LLM parse the problem context, it provides context signals at the domain and task levels for retrieval, so as to accurately distinguish different formal variants of concepts.
>
> Formal symbols require high accuracy. The mathematical formalization requirements are completely consistent with the symbolic system of the underlying theorem prover. The content retrieved by general RAG is mostly factual text, while mathematical formalization requires compilable accurate symbol identifiers. It is easy to return definitions with symbol mismatch or outdated version only by semantic similarity, which directly leads to compilation failure. We use dual-path hybrid retrieval, which combines symbol-level keyword matching with semantic vector retrieval, ensuring that the returned results are both semantically relevant and symbolically strictly consistent with Mathlib.
>
> The problem of conceptual implicit expression. The core mathematical concepts of many mathematical problems (such as combinatorial problems) are implicit in the description, lacking explicit conceptual keywords. General RAG searches based on the keywords of the original text, which is completely ineffective for such problems and cannot establish a bridge from natural language description to formal concepts. Aiming at the characteristics of implicit expression of concepts in mathematical problems, we introduce a problem rewriting module based on LLM, which transforms informal descriptions into expressions with explicit mathematical structures, so as to realize the accurate extraction of core concepts.

---

> ### Author Response · Authors · 2025-11-28
> **Response to Reviewer UEGf**
>
> **Weakness2: The experimental comparison focuses mainly on RAG-style retrievers, without including other autoformalization-specific retrieval systems.**
>
> We thank the reviewer for highlighting the importance of comparing against autoformalization-specific retrieval systems. We fully agree that such comparisons provide a more complete assessment of our method. As retrieval-augmented approaches for automated formalization remain in an early stage, publicly available systems designed specifically for this task are still highly limited. Consequently, our original submission primarily focused on representative RAG-style retrievers widely used in related domains.
>
> Following the reviewer’s suggestion, we have expanded our evaluation to include Dependency Retrieval (DR) \[1\], one of the few open-source autoformalization-oriented retrieval models. DR incorieves an ACS of 1.93 and a HitRate@3 of 42.9%, compared to DR’s 1.26 and 28.5%. These improvements highlight the advantage of CRAMF’s concept-driven retrieval design, which integrates contextual query augmentation and hybrid retrieval to better handle conceptual polymorporates a dependency-based retrieval mechanism and thus serves as a meaningful point of comparison for assessing task-specific retrieval effectiveness.
>
> We evaluated DR using the same metrics defined in our paper (Section 3.1.3): Average Contribution Score (ACS) and Relevant Definition Hit Rate (HitRate@3). The results (shown in Tables 1 and 2 below) indicate that CRAMF consistently outperforms DR on every dataset and metric. For instance, on AdvancedMath, CRAMF achphism and semantic gaps.
>
> Table 1: Average Contribution Scores (ACS) of retrieval methods on three datasets.
>
> |     |     |     |     |
> | --- | --- | --- | --- |
> | Method | MiniF2F | ProofNet | AdvancedMath |
> | Dependency Retrieval (DR) | 1.39 | 1.47 | 1.26 |
> | CRAMF | 2.07 | 2.14 | 1.93 |
>
> Table 2: Top-3 hit rate of relevant definitions (HitRate@3) for each retrieval framework across three datasets.
>
> |     |     |     |     |
> | --- | --- | --- | --- |
> | Method | MiniF2F | ProofNet | AdvancedMath |
> | Dependency Retrieval (DR) | 32.6% | 33.8% | 28.5% |
> | CRAMF | 44.2% | 50.6% | 42.9% |
>
> We also note that DR dependency-based retrieval mechanism mainly relies on explicit structural associations in formal libraries. Although this design can capture topological connections between definitions, it is intrinsically limited by surface grammatical structures and ignores deep semantic similarities at the conceptual level. This structure-oriented retrieval method leads to its rigidity in the face of conceptual polymorphism or cross-domain mathematical concepts. In contrast, CRAMF explicitly models the mapping relationship between natural language and formal definitions through the structured concept definition knowledge base, effectively taking into account semantic relevance and symbolic accuracy. At the same time, CRAMF introduces a context-aware query enhancement module, which analyzes concept semantics and reconstructs context through LLM, so that it can accurately identify the abstraction level and domain background of polysemous concepts, and achieve accurate definition and positioning in complex mathematical problems.
>
> We appreciate the reviewer’s suggestion, which allowed us to strengthen our experimental analysis. We will incorporate this comparison, along with the corresponding discussion, into the revised version of the paper to provide a more comprehensive evaluation of retrieval systems tailored to autoformalization.
>
>
> \[1\] Liu, Qi, et al. "Rethinking and improving autoformalization: towards a faithful metric and a dependency retrieval-based approach." The Thirteenth International Conference on Learning Representations. 2025.

---

> ### Author Response · Authors · 2025-11-28
> **Response to Reviewer UEGf**
>
> **Weakness3: The theoretical analysis of hybrid retrieval module to improve semantic consistency is lacking.**
>
> We have a theoretical basis for why hybrid retrieval strategy improves the semantic gap. First, our core motivation for designing dual-path hybrid retrieval stems from the unique need that semantic gap and symbolic accuracy must be addressed simultaneously in automatic formalization tasks. CRAMF captures similarities between natural language queries and formalized concepts in abstract mathematical meanings through concept-level semantic similarity retrieval (e.g., mapping "limits" to Filter.Tendsto). Symbol-level keyword matching ensures symbol accuracy. It can directly match the defined identifiers in Mathlib, effectively preventing compilation errors caused by "fabricating" non-existent symbols in the model.
>
> At the same time, in Section 3.3 (RQ2) of the original article, we have provided strong, final task-oriented evidence for the effectiveness of the retriever and knowledge base through the two metrics ACS and HitRate @ 3. Especially on ProofNet, the HitRate @ 3 is as high as 50.6%, which shows that for more than half of the questions, the system can find the "golden definition" that is precisely used in the final formalization code in the first three results. This speaks to the excellent coverage and ranking capabilities of our knowledge base. Meanwhile, the significantly higher ACS values than the baseline indicate that the retrieved definitions are semantically highly related to the problem. This indirectly but strongly proves that the mapping from informal problems to formal definitions is highly accurate and the embodiment of high precision.

---

### Official Review · Reviewer_bRZa · 2025-11-04

**Soundness:** 3
**Presentation:** 3
**Contribution:** 3
**Rating:** 6
**Confidence:** 3

**Summary:**

This paper proposes CRAMF, a retrieval-augmented framework that enhances LLM based automated formalization in theorem provers such as Lean 4.
The method builds a structured concept-definition knowledge base from Mathlib4 by aligning formal definitions with natural language expressions through reverse translation, concept extraction, and embedding via MathBERT. CRAMF retrieves relevant formal definitions for a given theorem through a dual pathway hybrid retrieval.

**Strengths:**

The motivation of model hallucination and conceptual polymorphism is sound, and the solution through the integration of retrieval and concept-level grounding is persuasive.
The ontology-based KB design is novel in the formal reasoning domain and technically sound
Experiments are comprehensive, covering multiple datasets (miniF2F, ProofNet), models (GPT-4o, DeepSeek-V3, Herald-7B), and ablations, and the newly released AdvancedMath benchmark adds value to the community.
There are consistent gains across heterogeneous base models

**Weaknesses:**

1. The conceptual extraction and rewriting modules rely heavily on opaque LLM prompting, I'm curious about the reproducibility and generalization.
2. comparison to premise/statement retrieval specialized for Lean is missing, which makes it hard to situate CRAMF against strongest formal-specific retrievers.

**Questions:**

1. could the authors report latency and context budget
2. How sensitive are results to KB noise?

---

> ### Author Response · Authors · 2025-11-28
> **Response to Reviewer bRZa**
>
> Thank you to the reviewers for their careful review and valuable opinions on our work. We are very pleased that you recognize the motivation of this paper in solving the problems of model illusion and conceptual polymorphism, the innovation of ontology-based knowledge base design, and the comprehensiveness of experimental setting. In response to your questions about system latency, context budget, knowledge base noise sensitivity, comparison with dedicated retrievers, etc., we have conducted detailed responses and analysis one by one below, and will supplement experiments and discussions accordingly in the revised version of the paper.
>
> **Weakness1: The concept extraction and rewriting module relies too much on LLM hints and reproducibility issues.**
>
> We provide detailed prompt words (Appendix C) for all LLMs used in the experiment in the appendix, and submit relevant codes in the attachment materials, which has ensured the reproducibility of the method to the greatest extent.
>
> **Weakness2: CRAMF lacks lean-related premise/statement retriever contrasts.**
>
> We sincerely thank the reviewer for raising this important point regarding comparisons with specialized formal retrievers. In response, we have now included a comprehensive comparison against two Lean-specific retrieval engines—LeanSearch \[1\] and LeanExplore \[2\]—which are explicitly designed for semantic search over Lean 4 declarations and Mathlib4. As shown in the updated Tables 1 and 2, CRAMF consistently outperforms both systems in Average Contribution Score (ACS) and HitRate@3, demonstrating its stronger capability in retrieving accurate and relevant definitions in real formalization scenarios. Relevant results have been added to the paper.
>
> Table 1: Average contribution scores (ACS) of retrieval methods on three datasets.
>
> |     |     |     |     |
> | --- | --- | --- | --- |
> | Method | MiniF2F | ProofNet | AdvancedMath |
> | LeanSearch | 1.58 | 1.63 | 1.56 |
> | LeanExplore | 1.42 | 1.51 | 1.35 |
> | CRAMF | 2.07 | 2.14 | 1.93 |
>
>
> Table 2: Top-3 hit rate of relevant definitions (HitRate@3) for each retrieval framework across three datasets.
>
> |     |     |     |     |
> | --- | --- | --- | --- |
> | Method | MiniF2F | ProofNet | AdvancedMath |
> | LeanSearch | 36.8% | 42.3% | 35.1% |
> | LeanExplore | 31.5% | 37.7% | 23.8% |
> | CRAMF | 44.2% | 50.6% | 42.9% |
>
>
> [1]  Gao, Guoxiong, et al. "A semantic search engine for Mathlib4." arXiv preprint arXiv:2403.13310 (2024).
>
> [2]  Asher, Justin. "LeanExplore: A search engine for Lean 4 declarations." arXiv preprint arXiv:2506.11085 (2025).

---

> ### Author Response · Authors · 2025-11-28
> **Response to Reviewer bRZa**
>
> **Question1: System latency and context budget report.**
>
> We thank the reviewers for their important questions regarding system latency and contextual budgeting. Here's our detailed analysis of the CRAMF framework on both fronts. Relevant content will be added to the appendix.
>
> **Delay analysis of each module of the system**
>
> We conducted a fine-grained breakdown of the end-to-end latency of CRAMF under a batch size of 1, using a hardware setup of 4 × NVIDIA A800 80GB GPUs. The results are summarized in the Table 1. We report the mean, standard deviation, minimum, and maximum latency of each CRAMF module to comprehensively characterize their delay distribution and performance variability.
>
> Table1: Latency Breakdown of Individual Modules in the CRAMF Framework
>
> |     |     |     |     |     |
> | --- | --- | --- | --- | --- |
> | **Module** | **Mean(ms)** | **Std(ms)** | **Min(ms)** | **Max(ms)** |
> | **Concept Extraction & Parsing** | 2261.63 | 181.26 | 2002.48 | 2903.49 |
> | **Symbol-Level Keyword Matching** | 124.177 | 35.455 | 103.197 | 191.172 |
> | **Semantic Similarity Retrieval** | 363.157 | 20.141 | 326.893 | 407.951 |
> | **Reranking** | 14.40 | 47.43 | 7.69 | 485.81 |
> | **Generation – Herald-7B** | 1752.107 | 295.404 | 1070.582 | 2220.342 |
> | **Generation – GPT-4o** | 9621.919 | 1080.509 | 8414.496 | 11440.511 |
> | **Generation – Kimina-7B** | 7633.742 | 5234.665 | 4118.432 | 19675.501 |
> | **Generation – DeepSeek-V3** | 6903.244 | 444.055 | 6443.822 | 7998.044 |
>
> Table 2: End-to-End Latency of CRAMF Integrated with Different Generation Models
>
> |     |     |
> | --- | --- |
> | **Model** | **End-to-End Latency (ms)** |
> | **Herald-7B** | **4515.48** |
> | **GPT-4o** | **12385.29** |
> | **Kimina-7B** | **10397.11** |
> | **DeepSeek-V3** | &nbsp;**9666.61** |
>
> **Context Budget Analysis**
>
> We have performed a detailed analysis of token usage for all models and datasets before and after using CRAMF enhancement, as shown in Table 3. The results show that although CRAMF does increase the context length, the maximum token usage is within 4.3 k, which is completely within the context window bearing range of the current LLMs.
>
> Table3: Token usage for each model and dataset before and after using CRAMF enhancement.
>
> |     |     |     |     |
> | --- | --- | --- | --- |
> | Dataset | Condition | Min Tokens | Max Tokens |
> | AdvancedMath | Base | 140 | 386 |
> |     | +CRAMF | 253 | 4,256 |
> | MiniF2F | Base | 136 | 299 |
> |     | +CRAMF | 222 | 1,310 |
> | Proofnet | Base | 131 | 281 |
> |     | +CRAMF | 401 | 4,085 |
>
> Through reasonable modular design, CRAMF achieves a significant improvement in formal accuracy and achieves a good balance of efficiency and effectiveness under the premise of introducing limited time delay and controllable context overhead.
>
> **Question2: The sensitivity of the automatic formalization results of the model to the noise in the knowledge base.**
>
> We performed controlled noise-injection experiments on MiniF2F using Herald-7B. Retrieved definitions were grouped by relevance deciles to simulate increasing semantic noise. We still use the evaluation indicators in the paper-Compilation Pass Rate@10 (CPR @ 10) and Formalization Accuracy Rate@10 (FAR @10). The experimental results are as follows:
>
> Table 4: Compilation Pass Rate@10 and Formalization Accuracy Rate@10 of MiniF2F data under different set definition noise disturbances.
>
> |     |     |     |
> | --- | --- | --- |
> | Retrieved Definition Group | Compilation Pass Rate@10 | Formalization Accuracy Rate @10 |
> | top1-10 | 93.0% | 62.3% |
> | top11-20 | 87.0% | 53.7% |
> | top21-30 | 82.4% | 47.5% |
> | top31-40 | 77.5% | 42.2% |
>
> The experimental results show that the formal performance of the model decreases significantly with the decrease of the relevance of retrieval definitions. Performance is best when the most relevant top1-10 definitions are used; When using the top31-40 definition with the lowest correlation, CPR and FAR decreased by about 15.5 and 19.4 percentage points, respectively, and the performance was even lower than that of the baseline model without retrieval enhancement. This clearly confirms that retrieval noise can have a significant negative impact on the translation process. However, in the CRAMF standard process, we prioritize the Top-3 high-correlation definitions through the dual-path mixed retrieval and reordering module (as described in Section 2.3.2 of the paper), which greatly reduces the probability of noise injection, thereby effectively guaranteeing the context quality and alleviating noise sensitivity issues.

---

### Meta-Review · Area_Chair_M1ZN · 2026-01-04

**Summary:**

The paper introduces a retrieval-augmented auto-formalization framework that retrieves Mathlib4-related definitions to reduce hallucinations and semantic gaps. All reviewers agree that the method is sound and that the performance gains are consistent.

The main concern raised in the reviews is the lack of certain evaluation settings. Reviewer 1 and Reviewer 4 mention the absence of comparisons with Lean-related retrievers, Reviewer 2 points out missing auto-formalization–specific retrieval systems, and Reviewer 3 suggests evaluating stronger and more recent LLMs.

Another significant concern is novelty. Reviewers 2 and 3 view the method as an adaptation of existing RAG frameworks to the auto-formalization domain. Reviewer 3 further notes that components such as reverse translation and query enhancement have been proposed in prior works.

Additional concerns include reproducibility, generalization, the lack of formal justification for the hybrid retrieval pipeline, and insufficient explanation of the AdvancedMath dataset.

The authors’ rebuttal provides clarifications and additional results that address all major concerns. It includes new comparisons with Lean-related retrievers, recent auto-formalization–specific retrieval systems, and stronger LLMs (including Gemini 2.5 Flash, Claude 4.1, and Goedel-Formalizer-V2-8B), all showing consistent improvements. The rebuttal also clarifies the differences between prior Lean-related methods and RAG-style approaches with additional results. Lastly, the rebuttal provides more analysis on reproducibility, formal justification, latency, and sensitivity to knowledge base noise.

Overall, the paper demonstrates solid empirical performance, and the rebuttal convincingly addresses the majority of the reviewers’ concerns. My recommendation is accept.

**Reviewer Concerns:**

The rebuttal addresses most of the reviewers’ concerns, including comparisons with prior methods, evaluation settings, and automatic evaluation metrics.

Reviewer 1’s concerns regarding the reproducibility and generalization of the conceptual extraction and rewriting modules might only partially be addressed: while the rebuttal clarifies reproducibility, it does not sufficiently discuss generalization.

Reviewer 2’s concern about the lack of formal justification for the hybrid retrieval pipeline might not be fully resolved. The rebuttal provides an intuitive explanation—that the hybrid pipeline addresses the two challenges: semantic gap and symbolic accuracy—which is reasonable but is not formal.

**Reviewer Scores:**

Reviewer 1, Reviewer 3, and Reviewer 4 are likely to raise their score, as the rebuttal provides comprehensive new results and clarifications.

---

### Decision · Program_Chairs · 2026-01-26

Accept (Poster)